# Sustainable Hybrid Manufacturing of AlSi5 Alloy Turbine Blade Prototype by Robotic Direct Energy Layered Deposition and Subsequent Milling: An Alternative to Selective Laser Melting?

**DOI:** 10.3390/ma15238631

**Published:** 2022-12-03

**Authors:** Jaka Dugar, Awais Ikram, Damjan Klobčar, Franci Pušavec

**Affiliations:** Faculty of Mechanical Engineering, University of Ljubljana, Aškerčeva cesta 6, SI-1000 Ljubljana, Slovenia

**Keywords:** computer-aided manufacturing (CAM), computer-aided design (CAD), hybrid processing, robotic milling, design for additive manufacturing (DFAM), wire arc additive manufacturing (WAAM)

## Abstract

Additive technologies enable the flexible production through scalable layer-by-layer fabrication of simple to intricate geometries. The existing 3D-printing technologies that use powders are often slow with controlling parameters that are difficult to optimize, restricted product sizes, and are relatively expensive (in terms of feedstock and processing). This paper presents the development of an alternative approach consisting of a CAD/CAM + combined wire arc additive-manufacturing (WAAM) hybrid process utilizing the robotic MIG-based weld surfacing and milling of the AlSi5 aluminum alloy, which achieves sustainably high productivity via structural alloys. The feasibility of this hybrid approach was analyzed on a representative turbine blade piece. SprutCAM suite was utilized to identify the hybrid-manufacturing parameters and virtually simulate the processes. This research provides comprehensive experimental data on the optimization of cold metal transfer (CMT)–WAAM parameters such as the welding speed, current/voltage, wire feed rate, wall thickness, torch inclination angle (shift/tilt comparison), and deposit height. The multi-axes tool orientation and robotic milling strategies, i.e., (a) the side surface from rotational one-way bottom-up and (b) the top surface in a rectangular orientation, were tested in virtual CAM environments and then adopted during the prototype fabrication to minimize the total fabrication time. The effect of several machining parameters and robotic stiffness (during WAAM + milling) were also investigated. The mean deviation for the test piece’s tolerance between the virtual processing and experimental fabrication was −0.76 mm (approx.) at a standard deviation of 0.22 mm assessed by 3D scanning. The surface roughness definition Sa in the final WAAM pass corresponds to 36 µm, which was lowered to 14.3 µm after milling, thus demonstrating a 55% improvement through the robotic comminution. The tensile testing at 0° and 90° orientations reported fracture strengths of 159 and 161.3 MPa, respectively, while the yield stress and reduced longitudinal (0°) elongations implied marginally better toughness along the WAAM deposition axes. The process sustainability factors of hybrid production were compared with Selective Laser Melting (SLM) in terms of the part size freedom, processing costs, and fabrication time with respect to tight design tolerances. The results deduced that this alternative hybrid-processing approach enables an economically viable, resource/energy feasible, and time-efficient method for the production of complex parts in contrast to the conventional additive technologies, i.e., SLM.

## 1. Introduction

The rapid advancement in additive manufacturing (AM) technologies within recent years has significantly changed the basic philosophies behind product design by incorporating the fabrication of complex geometries, which cannot be achieved with any single conventional process. Additive technologies have attracted a great deal of attention for applications that use moldable, ductile materials aiming towards potentially fulfilling production challenges by decreasing the design-to-manufacture time through replacing serial production processes with a single-step process approach. The AM methods achieve potential savings with respect to the raw resources, since only the material needed to fabricate the desired part is utilized; thus, the waste can be minimized, and leaner production goals can be realized. Moreover, AM stimulates the possibility of manufacturing cost-effective customized products and hybrid materials for obtaining specific functional properties that might be otherwise unachievable via a single conventional method [1]. Due to the increased demand for metallic prototypes and components, various metal-printing technologies have also been developed, such as selective laser re-melting (SLM), laser surfacing (LS), electron beam powder and wire re-melting (EBM), and metal inert gas (MIG)/metal active gas (MAG) or wire arc additive manufacturing (WAAM) [2,3,4,5,6,7]. These processes utilize a similar concept of melting the metallic precursors in the form of wire or powder directly with the arc or concerted energy beam. Generally, the parts or surfaces fabricated by arc-welding methods may additionally require post-surface processing and conventional machining to cover for the roughness and heterogeneous distribution of the material at the surfaces. Material losses during these post-finishing processes also contribute to the resource and re-work costs in the conventional welding or arc deposition methods [2,3,8].

WAAM can be classed as a direct energy deposition (DED) technique according to ASTM F2792-12a and, technically, it follows the principle of electric arc sequences (as a heat source) employed on a wire-based feedstock material [9]. Geng et al. [10] calculated a mathematical model for the wire-flying distance (arc zone) to compensate for displacement (start position offset) during gas tungsten arc welding (GTAW), such that a small feed angle of 10° and displacement of 3.5 mm ensures smoothly deposited layers. The classification of surface waviness during AM of the 5A06 aluminum alloy with GTAW has also been linked to current levels such that at low heat levels there is no weld penetration (bead formation). Whereas for a high heat input, the convex surface of the weld bead is remelted, implying that (a) wetting, and (b) the spreading of molten wire on the convex surface and the spherical cap’s remelting, are the two underlying mechanisms governing the formation of surface waviness (a curved pattern). These results thus propose that the WAAM processes are governed by two distinct forming mechanisms, i.e., wetting and remelting [11]. 

Aluminum alloys especially have abundant applications in aerospace and transport industries, and their utility is extensively growing in the light and heavy automotive sectors due to the utilization of multiple different parts for a variety of needs, e.g., sheet metal components, structures, etc. [1,5]. The high-strength cast Al-Si alloys typically contain a coarse, acicular silicon-rich secondary phase along with smaller Mg-containing precipitates. The larger Si-rich precipitates can contribute to ductility reduction and thus necessitate refinement for achieving the alloy’s high-strength attributes [4]. The replacement of steel by tough Al alloys in automobiles confers a ~30% reduction in energy utilization and weight along with better recyclability and corrosion resistance, leading to a substantial reduction in CO2 emissions [2]. However, these Al alloys necessitate stricter production quality control, high electrical conductivity, short fatigue lifecycles, and difficult welding/joining processes due to the development of thermally induced residual stresses. In turn, these mechanisms may lead to structural distortion, restricted rigidity, porosity, joint softening, fatigue-induced wear, a low coefficient of strength, and, eventually, intergranular cracking [5,12,13]. Gas pores and coarse grains typically provide poorer mechanical properties in additively manufactured AlSi5 alloys, so Wang et al. [14] identified low arc current and low pulse frequency ranges that could satisfy fine-grained uniform microstructures. The intrinsic formation of a surface oxide layer on Al alloys, which have a significantly higher melting temperature and corrosion inhibition characteristics than the pure aluminum itself, increases the difficulty of welding because this oxide phase remains stable even during the crucible/arc melting of the metal [15]. For the adequate welding of Al alloys, it is imperative to rupture this highly stable Al_2_O_3_ refractory oxide layer from the surface and key properties, e.g., the high thermal expansion coefficient and thermal conductivity, great degree of shrinkage during solidification in a broad temperature range, and the high solubility of hydrogen must also be compensated [16]. Gu et al. [17] reasoned that the pre-existence of contaminants in feeding wire (poorer quality) can lead to hydrogen uptake and hot cracking in Al alloys due to high thermal stresses and shrinkage during solidification, especially in the 2xxx and 7xxx (aerospace) series. During MIG-WAAM (wire arc additive manufacturing), the shielding gas covering the consumable electrode helps eradicate the oxide from the melt pool; however, thermally induced residual stresses may result in a higher susceptibility to undercuts, distortions, or hot cracking along the heat-affected zone (HAZ) [6,18]. Still, with the correct WAAM surfacing parameters for Al alloys, e.g., the correct joint preparation, part fixation angles, gap reduction, degree of stiffness, shielding-gas flow rate, HAZ temperatures, thermal sequences, etc., hyperstatic structural joints can form [4,13]. Wu et al. [16] reported that high-angle wire feeding at 60° can optimally lead to consistent deposition, that can be attributed to the arc’s uniform temperature distribution for droplet formation and molten pool solidification. A low feeding angle (30–50°) caused the termination of back feeding, while the higher angle > 70° led to the littering of the droplets due to the increase in the side arc’s electromagnetic force FX. Generally, these welding optimizations are accommodated manually; thus, the automation of the process of joining the hyperstatic structures with Al alloys is considered challenging [2,19].

Consequently, more recent optimizations of WAAM process for Al alloys have been vindicated in several articles. Horgar et al. [20] reported the use of GMAW with a wire composed of the AA5183 Al alloy on an AA6082 T6 support base and identified intergranular hot cracking within the multilayers near the fusion boundary region due to reheated weldment from subsequent passes. However, the dominantly isotropic microstructure, which resulted in a 293 MPa tensile strength value, delivered comparably higher strengths and hardness values over the commercial alloys. Ismail et al. [21] recently explored the possibility of developing a WAAM EN-AW6016 (6xxx series) Al alloy with less than 1% porosity and excellent fusion; however, embrittlement and mechanical properties lower than the T4 state of the as-deposited alloy were reported. Köhler et al. [22] fabricated linear walls of Al-4046 and Al-5356 alloys by WAAM and explained that the solidification and setting responses significantly impact the surface waviness; thus, the increased arc lengths and energy pulse created higher dynamic forces, which affect droplet morphology and deposition accuracy. Typically, residual stresses also form in thin-walled structures, i.e., the bottom substrate and first layer experience tension, while the top and the layer beneath it experience compression. Gu et al. [23] utilized interlayer rolling with loads from 15–45 kN for each subsequent WAAM pass on 5087-grade Al alloy, which resulted in a simultaneous increase in yield strength (YS), ultimate tensile strength (UTS), and microhardness. Moreover, this sequential layer rolling at 45 kN caused the reformation, pore closure, and grain refinement of the microstructure with an ~82.2% areal fraction of fine grains (<10 µm) under the mechanisms from Hall–Petch model, i.e., deformation-induced-high-density dislocations and a substructure generation. The two strengthening effects for WAAM Al-6.3Cu alloys involving interlayer cold working and post deposition thermal treatments have also been investigated, with the former returning a 314 MPa UTS when rolled at 45 kN and the latter enhancing the UTS to 450 MPa after T6 treatment (in both cases with and without rolling) [24]. Gu et al. [25] claimed to have greatly reduced the pores larger than 5 µm in the WAAM 2319 and 5087 series Al alloy by subsequent 30–45 kN interlayer cold rolling following molten weld deposition. Without rolling, a larger pore areal fraction was observed, which was significantly reduced as pores larger than 5 µm were effectively mitigated by 45 N interlayer rolling. The applied pressure had its obvious benefits in terms of atomic hydrogen absorption, porosity reduction, and microstructure refinement, which improve the mechanical properties to a degree that is on par with the machined Al-alloy billet. Nevertheless, interlayer rolling cannot deform the solidified weldment beads, which may lead to cracking and defects in the HAZ region; so, essentially, the welding parameters and thermal treatments assert their priority [9]. 

Wang et al. [26] confirmed grain refinement with Al_3_Ti phase formation in Al5356 series thin-walled WAAM components after the addition of titanium powder. The interlayer microstructure refinement corresponded to a change from columnar to equiaxed grains, which, subsequently, also augmented the UTS and microhardness along with the isotropic elongation characteristics. Sales et al. [27] utilized the potential of adding 0.2–0.5 wt.% scandium to the AA5183 Al alloy over an AA5083 plate, which yielded an improvement of a 60 MPa increment in YS and UTS in both the horizontal and vertical directions. Like Ti and Zr, scandium is also a grain refiner and forms ultrafine intermetallic precipitates of Al3Sc, which translate to higher strengths in these alloys. Morais et al. [28] evaluated the mechanical properties and microstructure of an Al–Zn–Mg–Cu alloy fabricated by WAAM and identified only minor defects, e.g., porosity and no sign of cracking or a lack of fusion. Moreover, the mechanical properties of the Al–Zn–Mg–Cu wire arc-fabricated alloy were reported to be better than commercial 7xxx series Al alloys. So, gone are the days when the workmanship-related precision and surface quality of products made by conventional welding routes were frequently poorer compared to beam/laser re-melting due to automation and sensitive arc deposition (micrometer-level precision) [19,29]. 

The combination of layered direct energy deposition (DED) and material removal by milling (subtractive processing) offers potential advantages over the conventional manufacturing approaches and these individual processes. Computer-aided design (CAD) coupled with computer-aided manufacturing (CAM) and subsequent machining offers a comprehensive solution for workpiece conceptualization (planning), parametric adjustment (design), and quality-assured (QA) processing. Automation and modularization are inarguably the most effective methods for increasing the competitiveness, productivity, and manufacturing flexibility suited to complex parts via bridging perks such as standardization and the restructuring of production philosophy by further integrating the design of experiments (DoE) with CAD/CAM [5]. By fabricating a near-net product, it is possible to reduce the amount of material loss, which is especially suitable for expensive and hard-to-machine materials. In turn, this reduces tool wear, the time needed to produce parts, resource material consumption, process sustainability, and end product costs [1,6]. The subsequent advantage of hybrid processing also suggests the possibility of forming intricate parts/sections (e.g., deep and narrow slots, cooling channels, cavities, etc.). These would otherwise be difficult to make with the conventional metallurgical processes and especially in larger batches, thereby suggesting affordable and sustainable manufacturing. With the design freedom accessible via implementing a hybrid approach, there is the potential to considerably augment the efficiency, productivity, and functionality of existing designs and integrate part complexity, e.g., internal orifices, channels, or structures, such that the overall design is not sensitive to cost [1,7]. Moreover, this hybrid WAAM + milling approach can be tailored for prototype development by the scalable co-deposition of different metallic materials and alloys in complex shapes, depending on the functional requirements [30]. The amalgamation of WAAM automation and robotics are suitable for the dynamic production environment, a proposed solution that delivers the best ‘cost per unit’ productivity [5,7]. Thus, both these technologies, i.e., WAAM and milling, have their advantages and disadvantages. The use of a hybrid AM route in lieu of only one of these aforementioned technologies promises the sustainable manufacture of challenging-to-automate, larger-sized structures of high-strength Al alloys on an industrially viable scale.

The scheme of this article is as follows: Section 2 interprets our achieved contributions to the field and the explanation of the concept of hybrid manufacturing. Later, the design of experiments for geometric specifications with CAD is detailed and followed by an in-depth review of CAM parameters for the virtual simulation of a robotic WAAM process and milling. In line with CAM virtual processing, the Section 5 describes the robotic WAAM and milling parameters for prototype fabrication. The Section 6 evaluates the differences in a fabricated AlSi5 alloy workpiece under several WAAM parameters and consequent robotic milling to consolidate dimensional precision. Consequently, these fabricated parts were examined by 3D scanning for their geometric conformity, and performance evaluation in terms of mechanical testing, surface roughness, and microstructural examination. At the end, sustainability indictors in terms of processing time (efficiency), costs, and energy utilization are analyzed between hybrid manufacturing and selective laser melting. The research provides guidelines and suggests that implementing the optimization of the CAD/CAM suite with robotic WAAM + mechanical communition as a collective fabrication route offers better viability than the costly and more time-consuming SLM methods in the case of free-forming complex metallic parts.

## 2. Hybrid Production

The convergence of contemporary CAD/CAM-assisted comminution processes with automation and robotics within manufacturing (repetitive, batch, and continuous) operations has delivered excellent surface quality with accurate geometrical tolerances at high machining speeds. Robot-assisted milling offers benefits regarding factors such as (a) precision—the cutting tools provide greater accuracy than current AM methods; (b) finishing—the possibility of achieving smoother surfaces (which, using the current AM methods, result in coarse layer sections and top surfaces); (c) mass production—faster and cheaper work with large quantities of identical pieces; and (d) choice of materials—the potential to process different types of materials and with higher degrees of freedom compared to simple AM methods, which are based on the filaments of only a specific type of material type [1,2,7]. Nevertheless, forming pieces with complex geometry is more difficult with modern cutting practices because not all surfaces can be machined with super hard high-speed tools either. The wire arc additive manufacturing (WAAM) method, in comparison to laser-based AM methods, offers the advantage of large-scale material applications for fabricating larger workpieces and retaining better surfaces due to the avoidance of thermally induced (irradiance) splatter of powders [18]. The workpiece size is more important in specific materials, e.g., Ti alloys, which necessitate the use of an inert, protective atmosphere, so the concurrent laser-based AM setups have limited fabrication beds or printing chambers. Thus, a user can navigate with freedom in favor of wire arc deposition process for larger-size aluminum alloy parts [5,7]. In addition, the wire feed mechanism supplies the filler wire atop the welding zone at a steady speed without interruption, so deposition is fast, and the quality of the weld is better than with manual arc welding [2]. The reverse movement of the wire improves the separation mechanism of the molten droplet during the short-circuit current [2,12]. The microprocessor synergistically automates the current source and enforces a low value, thus regulating the molten alloy transfer without spraying, splashing, or wider-scale splattering. 

To prevent weldment spillage and interrupted wire supply to the welding zone in the event of a short circuit, the cold metal transfer (CMT) arc deposition (WAAM) technique permits much lower heat input than the traditional MIG process [31]. The CMT method offers digital control over the feed material, which is tethered to the welding current in a closed loop feedback system [32]. The CMT process is primarily intended for welding very thin sheets or for producing root welds, bridging the weldments in thicker sheets, and joining different types of material (brazing) [8]. In the CMT method, during the passage of the filler material onto the substrate, electric current practically does not flow, while the short-circuit current of classical arc welding is high [33]. Technically, when the welding arc burns out in the event of a short circuit, the decrease in the molten filler material is transferred to the substrate such that the surface tension of the molten metal supports the passage of additional filler material. Hence, the short-circuit current and the heat input are reduced with CMT, which, in turn, enables higher process sustainability [31]. Ortega et al. [34] studied the effect of CMT-WAAM parameters on Al5Si weld quality, inferring that highly precise 100-layer deposits can be made with a standard deviation of the wall width of ~0.3 mm. The variable-polarity cold metal transfer (VP-CMT)-resulted in to WAAM of an Al-6Mg alloy developed uniform equiaxed grains 20.6–28.5 µm with a random orientation, and, in turn, the UTS 333 MPa of the AM component was higher in comparison to the wrought alloy or other CMT modes yielding columnar-type grains [35]. The CMT-pulse-advanced (CMT-PADV) mode, according to Cong et al. [32], resulted in the lowest porosity for Al–Cu alloys. Importantly, CMT-PADV processing proved to fully eliminate the gas pores (an oxide-cleaning effect). Similarly, Fang et al. [33] verified the pore area percentage, aspect ratio, and their spatial distribution within the 2219-series aluminum-based workpiece to be lowest at 0.98% for the CMT-PADV process. Gu et al. [36] linked CMT-WAAM deposition parameters with the anisotropic mechanical properties of an Al–Cu–Mg alloy, affirming a microstructure composed of hierarchically dispersed dendrites as well as equiaxed and scarce columnar grains such that, with a follow-up heat treatment, the coarse secondary phases were refined by 95%.

On the contrary, the AM technologies such as SLM offer more flexibility in terms of geometric shapes since the deposition of material takes place in multilayers [37]. The additive manufacturing process can be relatively simple (less technical knowledge is required for its performance), and as a self-autonomous robot-controlled sequence, it may not require a physical human presence [3,8]. The existing expensive AM methods, which yield low surface roughness and poor control of asperities, can be associated with layer-by-layer deposition or re-melting methodologies [1]. Therefore, for the surface deposition method to compete with contemporary AM, the influencing parameters—such as the good re-melting of material, high deposition rate, flow viscosity, droplet morphology, melt temperature, scan rate/speed, and solidification control to a near net shape—are pivotal and must be tightly regulated [4]. Essentially, the preceding reports regarding the WAAM of Al alloys report only simple shapes [9], and a comparison with metal 3D printing techniques has not been made. The literature has proven that the CMT-WAAM technique is much more appropriate versus the conventional DED methods in terms of deposits’ optimization over multiple passes, refined microstructure (without hot cracking and porosity reduction), and mechanical strength levels, which are better than those of wrought-Al alloy [14,31,32,33,35]. Extensive WAAM multilayer control with autonomous robots has been meagerly detailed, which provides enough motivation to investigate the utility of anthropomorphic robot-clamping methods for a wire + arc torch setup in relation to obtaining complex shapes. The subtractive methods are important because of issues related to inappropriately solidified bead/layers with rough surface tolerances. The isolated application of surface machining over the finished parts cannot correct the interlayer macro defects (other than porosity or cracking, which are microdefects), but only the exterior dimensions, so resource wastage (material, energy, and time) is evident. In such cases, parts are commonly manufactured additively with wider tolerances, whereby the tandem subtractive processing of deposited surfaces creates a level playing field for the next passes and WAAM parameters or robotic automation that does not require additional compensation with respect to the parts’ geometry. So, in order to fabricate an internally hollow and arc-driven shape, e.g., a turbine blade workpiece consisting of multiple layers, we incorporated an inclusive robotic milling strategy for retaining the correct interpass characteristics, and the final geometry meets the tolerance designation derived from the CAD model. In this research, we justify how robotic CMT-WAAM + milling is suited for the fabrication of free-form workpieces in a more sustainable manner than SLM regarding cost, time, and energy consumption criteria.

In this study, the robotic cold metal transfer (CMT) MIG-WAAM (wire arc additive manufacturing) approach was employed for the deposition of an AlSi5 alloy (EN 18273: 4043 wire of 1.2 mm thickness) in the sustainable design of a turbine blade. After each weldment pass, simultaneous trimming of the surface layer was carried out by a robotic milling unit. The design of the experiments is defined in a flow chart illustrated in Figure 1. Due to the precision required for the deposition of this shape, the CMT WAAM was performed using a six-axis anthropomorphic industrial robot (model ABB IRB 140-6/0.8) with a rated power of 4.5 kW (ABB, Zurich, Switzerland), upon which a Robacta Drive burner torch was mounted, which was controlled by a Fronius TransPlus Synergic 3200 CMT R welding machine (3–320 A current output) (from Fronius International—INGVAR d.o.o. Ljubljana, Slovenia) and 99.98% purity Ar shielding gas at flow rate of 13 L/min, as shown in Figure 2a. Through the utilization of robots, we tuned the feed parameters, constant welding speed, and burner position (via RCU 5000i control unit and FlexPendent remote controller) (from Fronius International—INGVAR d.o.o. Ljubljana, Slovenia), which are important for the success and repeatability of the process. The robotic arm had a loading capacity of 5 kg, with a fifth axis reaching up to 810 mm, a proclaimed positional repeatability of 0.03 mm, a maximum tooling speed of 2.5 m/s with a tool acceleration of 20 m/s^2^, and a rated power of 4.5 kW. The parametric control enabled via CAD/CAM simulations helped classify different robotic welding and the milling parameters. It is important to understand that virtual processing with CAD/CAM is orders of magnitude more efficient than contemporary numerical methods because G-code can be derived directly from the CAM suite once the parameters match the designated specifications (product and process). These parameters were translated in prototype fabrication, and experimental results were later interconnected with the microstructural outcome and tensile tests. Lastly, generic calculations were estimated regarding the combined cost of processing such that the hybrid approach can be favored over different methods in an isolated mode, i.e., conventional machining, additive manufacturing, MIG welding and/or similar metallurgical casting, etc.

The milling procedure incorporated a KUKA KR 150-2 (KUKA Roboter GMBH, Gersthofen, Germany) robotic manipulator comprising a high load unit (110 kg peak weight) with a working space of 55 m^3^, positional accuracy of +/− 0.06 mm, maximum tooling rotation or spindle speed of 11,700 min^−1^, a water-cooled electronically-driven spindle power of 6.3 kW, as shown in Figure 3a. The spindle enables the option of adjusting the coolant temperature for the most optimal operations (represented in Figure 3b).

The physical properties and the chemical composition of the AlSi5 alloy filler (EN 18273: 4043) according to the standard reference from the manufacturer are given in Table 1 below:

A 6 mm diameter WAB312061 carbide ball-end cutter possessing two effective cutting edges, with a 5.5 mm cutting length (*L*_1_) and a flute length of *L*_2_ = 40 mm, which is typical for machining aluminum alloys, was utilized for the robot-assisted milling operations, as shown in Figure 4.

## 3. Materials, Methodology, and Design of Experiments

Separate robotic systems were utilized for the sustainable material deposition and cutting processes. According to the flowchart in Figure 1, the robotic machining procedure was designed indirectly using the software environment SprutCAM 11 (Pro/Robot Edition, Sprut Co., Devon, UK). The advantage of the SprutCAM program is its ease with respect to efficiently planning the movement of robots for the welding and milling processes. Since hybrid processing was performed on two different machines (from different manufacturers and specifications), two postprocessors were used to convert the planned movement and robot commands from the SprutCAM simulation environment into NC code, which was transferred via the machine controller. Some manual-surfacing NC adjustments of the process were also carried out in order to correct the imperfections in the postprocessor with respect to the welding parameters for each subsequent operation. The WAAM torch followed continuous rastering pattern for each layer, which was milled under the same path plan for surface tolerance control [38,39]. This is a more suitable approach over simple rastering, zigzag, or contour-based deposition because these strategies do not guarantee complete filling of a 2D geometry from contour pattern offset at curves or boundaries. Thus, continuous deposition for a layer is more suitable than other strategies. However, more complex path planning should incorporate hybrid approaches in continuous deposition in order to avoid leaving any section, curves, or boundaries with voids [40]. Ponche et al. [41] suggested incorporation of Design for Additive Manufacturing (DFAM) approach in the planning phase, as outlined in Figure 1, which encompasses design and manufacturing specialties per complex geometric models. Continuous or hybrid planning for tooling path thus advocated in these concurrent reports became our focus in the design phase [42]. Since the path plan was not very complex, layered processing in continuous mode was preferred over hybrid strategy during DFAM in order to maximize WAAM torch and milling tool’s mobility, with time and energy conservation in mind to compete against SLM.

The design process commenced with the import of the CAD model of the workpiece determining the shape and size of the parts, clamps, positioning, and orientation of the coordinate system of the workpiece according to the base coordinate starting point of the robot, along with the sequence of surface operations in the designated machining strategy. A selected test piece was made to verify the limitations of the combined processing method using robots in the frame of robotic wire surfacing, and 5-axis simultaneous milling-machining procedure was designed. To fulfill the conditions of the complex geometry of the test piece and optimization of hybrid processing, a model of gas turbine blade was selected, as shown in Figure 5 (dimensions in mm).

Typically, due to high temperature operations, special steel, nickel, and titanium alloys are used to make these turbine blades. However, for sustainable proof of concept, AlSi5 alloy was used instead in this study to demonstrate the feasibility of robotic hybrid fabrication. The production of a sample piece with the combined process of robotic surfacing and milling took place in several stages. The consumption of filler alloy was monitored by weighing the workpiece before and after subsequent welding operation with Mettler Toledo SB120001 laboratory balance (Mettler Toledo d.o.o, Ljubljana, Slovenia) with an accuracy of 0.1 g. The temperature of the workpiece was monitored by connecting K-type thermocouple to digital multimeter VOLTCRAFT M-3850 (Metex Corporation, Seoul, South Korea), which operates in the range of 40–1200 °C. After the final processing of the test piece, the measurement of the accuracy of production was performed on the Merlin Zeiss PRISMO Navigator coordinate-measuring device (ZEISS, Ljubljana, Slovenia) with a repeatability of 0.99 µm, touch force sensing at 200 µN, and linear error prediction range of 0.9 + L/350 µm, applied to the final model of the workpiece made in CAD software environment (Solidworks 2017 P4 ×64, November–December 2017). Similarly, the surface measurements of the geometry before and after machining were accomplished with a high-precision Alicona InfiniteFocus SL microscope (Bruker Alicona, Itasca, IL, USA), which enables the recording of a 3D model and property analysis in a dedicated software environment. The tensile tests were carried out on Zwick Z250 universal testing machine (Zwick Roell—Ebert d.o.o., Ljubljana, Slovenia) up to a maximum load force of 250 kN. WAAM samples were sliced in cross-sections for further microstructural investigation with low-speed saw-cutting machine Struers Discotom 5 (Struers LLC, Cleveland, OH, USA) [7]. Microstructure analysis was carried out on a measuring microscope for macro- and microstructure analysis with the Olympus BX61 image analysis system (Olympus—Labena d.o.o, Ljubljana, Slovenia), which also enables quantitative determination of the microstructure, size, shape, and distribution of various elements of the microstructure and porosity measurement.

The production of the sample piece by the combined process of robotic surfacing and milling took place in several stages. The aim here was to fabricate a hollow test piece using hybrid process that could be exploited to finish the product from the inside-out, which would have been otherwise impossible with the conventional machining. The strategy consisted of forming the workpiece in eight steps of 20 mm to a final height of 153 mm, as shown in Figure 6. The construction of sections with a height of 20 mm was opted due to the restriction of access with milling tools when machining the inside of the wall. The process required twisting between workbenches due to the use of separate robots for wire arc surfacing and milling process. During the work, it was observed that, in this phase of the layers’ development, due to the poor positional accuracy of the robot, the machining of the interior segments (inside surfaces) was not feasible; thus, the focus was shifted to exclusively the treatment of external surfaces.

## 4. Analysis of Arc Deposition Parameters (Planning Phase)

In the first phase of planning, the surfacing parameters of the test piece were selected according to a series of sample experiments performed at different combinations of the burner feed rate, welding current strength, and the corresponding welding voltage and wire feed rate [6]. With the selected parameters, the suitability of different strategies for surfacing structures with a slope were also tested. The deposition parameters (welding current strength, welding voltage, wire feed speed, and welding speed) were determined based on known results of existing research [7], which sought to achieve a constant weldment width > 5 mm and ensure the possibility of creating weld structures with an inclination.

The deposition parameters of the test piece were selected according to a series of arc depositions performed at different combinations of arc torch feed rates, welding current strengths, and the corresponding welding voltages and wire feed rates. With the optimum parameters, the deposition of the aluminum alloy AlSi5 with the burner feed rates of the arc discharge values were set between 6 mm/s and 10 mm/s. The range of welding currents to achieve the desired wall thickness was determined by depositing 50 mm long welds to produce walls consisting of 10 layers, as illustrated in Figure 7. By adjusting the welding current strength, the CMT process controller also automatically determines the wire feed speed and arc voltage [2,3]. The weld surfacing, i.e., WAAM, of 10 layers was sufficient for assessing the stability of the parameters, the walls’ corrugation (grooving), the remelting intensity, and, at the same time, the distance of the torch nozzle from the welding site (10–20 mm); due to the different layers, the height variation do not pose problems with respect to gas shielding. The welding torch was oriented perpendicularly to the substrate during deposition, while the feed direction was altered progressively with each layer (Figure 7).

The inter-pass temperature between the individual weld layers was initially maintained at 120 °C, as identified in Figure 8. By performing wire deposition experiments with different combinations of welding speeds, current strengths, arc voltages, and their related wire feed rates, the range of parameters suitable for achieving the optimal weld geometry for making a consolidated turbine blade was determined and listed in Table 2. The WAAM energy input *E* [kJ/mm], according to the EN-ISO 15614-1:2017 standard, can be calculated from simple relationship: *E* = (*U* × *I*)/*v* × 10^−3^, where *U* is the arc voltage [V], *I* is the arc current [A], and *v* is the welding speed [mm/s] [18]. However, it is necessary to pay attention to the amount of additional material, as the wire is fed in the deposition zone, which may alter with the strength/variation of the welding current.

Deposition with a higher welding speed of 10 mm/s proved to be less appropriate (samples A1–A4), as shown in Figure 9 with a red outline. The use of the CMT process at higher welding currents increases the energy input per unit length and the wire feed rate. At the same time, this causes rapid remelting of the AlSi5 alloy, so the wall thickness was chosen to be greater than 5 mm (a suitable parameter); however, a high feed rate does not allow for the stable deposition of the wire and, thus, a constant layer height was difficult to retain such that excessive ripples occurred in both the vertical and horizontal directions [29].

By reducing the welding current, the surface waviness was reduced, e.g., as in sample A3 vs. A1 or A2, but the material was poorly remelted; especially when in contact with the base material, the wall thickness was smaller, and thus less suitable for the continuation of WAAM. Decreasing the welding speed increases the energy input per unit length, which affects the higher remelting property of the AlSi5 material [18]. From Table 2, at a set current of 96 A and a wire feed rate of 8 mm/s, the wall width is larger (approx. 6.8 mm) and the layer height is lower (1.6 mm) than at a welding speed of 10 mm/s. Settling at both ends of the wall is more pronounced (depressed edges). By reducing the welding current of sample A3 from Figure 9, the construction of the wall was uniform, without settling of the ends, with minimal ripples in the vertical and horizontal directions. The layer thickness was 4.6 mm and the height 1.9 mm from Table 2. A further reduction in the welding current led to the construction of a 3.4 mm thin wall, which had poor remelting characteristics. With the excessive reduction in the welding current, the energy input per unit length is too small for the base material to melt and adhere to the deposited base layer [43] (i.e., sample A4 in Figure 9). The droplets in this case consist of hardened (solidified) filler material, which makes it unfeasible to build further layers on top as the mechanical integrity of the first deposit is poor [44].

Reducing the feed rate at the same welding current strength causes greater remelting (sample B1–B5 in Table 2), and at higher currents this resulted in a high degree of settling (depression) of the wall ends, as presented in Figure 9 (blue-outlined). With the welding speed decreased to 6 mm/s, the appropriate layer geometry could be achieved; however, there was some compromise at a decreased feed rate and current/voltage combination, i.e., C4. With an adequate combination of parameters in sample C3, a uniform 5.7 mm thick flat wall and 2.2 mm layer height, which is suitable for the purpose of the combined manufacturing process, could be processed, as shown in Figure 9 (green outline). It was also realized that at lower wire feed speeds, the corrugation of the wall is less pronounced due to the stabler arc generation and the more even melting of the material [12], and any minor deposition errors during melting can be mitigated by resurfacing in the next pass.

Importantly, regarding the B5 sample, marked in Figure 9 with a purple border, by suitably fine tuning the parameters of C3, the optimized deposition was achieved with a welding current strength of 73 A, a welding voltage of 12.1 V, and a wire feed speed of 4.1 m/min, translating to a flatter deposit without depressed edges. With lower welding current and voltage, the energy input was reduced; therefore, remelting was controlled to achieve a suitable wall thickness of 5.2 mm. This formed a lower layer height at 1.8 mm due to the lower wire feed speed, as indicated in Figure 9.

### 4.1. Adjusting the Shape of the Deposit

The welding program (in the CMT-WAAM system) allowed for the adjustment of parameters such as the ignition current for welding, the ignition/final current extent (duration), the current at the end of welding, and the interval between the transitions [43,45]. The parameters were adjusted so that the material at the beginning and at the end was not excessively melted. This ensured a uniform height and width of the deposit along its entire length of 173.1 mm.

### 4.2. Influence of Torch Placement in Horizontal Welding Position

To achieve a shape of the design workpiece with inclined surfaces, we were required to avoid supporting structures that needed to be removed by processing. Robot welding allows for any orientation of the welding torch, and this was used to weld the sloping walls in a horizontal welding position [12]. The flat walls were deposited at different angles in two ways, as illustrated in Figure 10, firstly, by orienting the burner perpendicular to the base plate by laterally shifting the layers as in Figure 10a, and secondly by orienting the burner in the direction of wall growth, as shown in Figure 10b.

The walls were deposited with a planned height of 50 mm and a length of 40 mm. Therefore, using the optimum parameters of the B5 sample, a welding current of 73 A with an arc voltage of 12.1 V, a wire feed speed of 4.1 m/min, and a burner feed of 8 mm/s were utilized. The deposits were made at inclination angles (*α*) of 75°, 60°, and 45° with respect to the base plate. The influence of gravity on the degree of melting during welding at different angles was also taken into consideration [7]. The height and width of the deposit and the deviation of the solidified melt in slope from these set angles were investigated and can be seen in Figure 11 below. In this first strategy with the welding torch oriented perpendicular to the base surface and the torch path shifted, a slope at which alloy deposition was not viable for achieving the desired geometry was at lower angles, e.g., at a 45° WAAM torch offset.

To achieve the oblique loading of the material, in the first case, we used the burner-shifting method, in which the WAAM torch was rastered longitudinally with respect to the previously welded layer for each subsequent layer (the welding torch was oriented perpendicular to the base surface). Based on the test results, it can be inferred that this strategy is suitable for the construction of walls with a small slope (up to 60° relative to the horizontal base). The measured angle of inclination of the walls corresponds to the set in the case of surfacing at an angle of 75° and 60°, and when the inclination of the wall was 45°, the wall settled poorly since the feed wire was in contact with the previous layer at the edge, and the remelted material flowed off the wall upon arc ignition. The height of the layers was uniform in the case of Figure 11a,b, in which the wall was relatively homogeneous but slightly wavy with a decrease in the tilt angle, while in case Figure 11c there was more remelting, which is reflected in the convexity of the surface and uneven height of the layers, mainly at the ends of the wall.

In the second instance, the strategy of tilting the WAAM torch in the direction of wall construction proved to be more stable, as shown in Figure 11a,b, since the surface is flat even when the wall is tilted at an angle of 45° in the case of Figure 11c and the height of the layers is uniform, so there was apparently no corrugation on the final layer. This strategy, therefore, incorporating the precise calibration of the welding torch, ensures the supply of filler material to the middle of the previous layer. The limitation of the walls’ deposition on a flat surface with higher slopes relates to the size of the shielding gas nozzle. The gas nozzle may hit the base plate during the deposition of the initial layers if the distance of the nozzle from the welding torch does not conform to 20 mm.

To weld the test piece, a base plate made of aluminum alloy 6061 having the dimensions 170 × 110 × 20 mm was utilized, which was placed 670 mm from the robot base on the *x*-axis and 540 mm on the *y*-axis to avoid singularities, as shown in Figure 12a through a virtual simulation. The test product was then welded to the base plate, on which we defined the coordinate origin of the workpiece in one of the corners. The same point was also used as the coordinate starting point for the milling process defined in Figure 12b, thus reducing the possibility of error due to the positioning of the workpiece on the clamping table.

When designing surface deposition, it was vital to ensure the accuracy of the position of the welding torch, as this guaranteed the accurate application of the material and reduced the necessary finishing volume by milling. Calibration was performed by moving the tip of the wire from the nozzle to a length of 15 mm, which certified the optimal distance from the workpiece to direct the shielding gas. With the tip calibration, the model then approached the reference point in three different tool orientations, which differed as much as possible in the rotation of the individual axes of the robot. The tethered robotic controller determines the position and orientation of the tool coordinate system from the captured data, and the data are used for planning in the software environment [42].

Machining design begins with importing the CAD model of the workpiece and clamps and placing them in the robot’s workspace. In this case, the workpiece was a roughly machined unit, on which we welded a new segment, and the clamping plate was included in the model due to possible collisions during the production of the actual piece. When conducting WAAM surfacing, the strategy of tilting the welding torch in the direction of the walls’ construction was utilized to achieve higher homogeneity of the walls and higher buildup of each layer, as well as ensuring the greater accuracy of weldments. A uniform depth of cut was also maintained during comminution to preserve structural homogeneity [6]. The most suitable strategy offered by the program was five-axis-machining approach, intended for surface treatment (5D surfacing), in which the tool (welding wire) follows the shape of the deposition surface with a definite offset, as shown in Figure 13. The workpiece with a final wall thickness of 3 mm, having a deviation from the surface of 1.5 mm from the inside of the wall, was designed. The movement of the tool took place at the middle section of the wall, and the addition of material for subsequent milling was the same on both sides (1–1.5 mm). In the production of the test piece, the exchange in counterclockwise directions of movement of the welding torch to weld each layer was adopted. The speed of fast movements was set to 17 mm/s, and the welding speed was controlled according to the selected welding parameters in Table 2 regarding sample B5 at 8 mm/s.

The WAAM torch was programmed to adopt a continuous rastering pattern for each layer in order to retain control over surface tolerances [38,39]. During the surface WAAM deposition within the virtual environment, the set parameters ensured the addition of 1–1.5 mm of material on each side of the wall. Mechanical processing was divided into two phases. The first was performed after the surfacing of a single segment with a height of 20 mm in two passes. After processing all the eight welded and machined segments, the second phase of processing the entire product was implemented using ball-end milling. So, in the first phase, the processed individual sections of 20 mm height were trimmed, with an additional 0.5 mm for finishing to reduce the cutting forces and vibrations during machining run.

A rotary machining strategy (around *z*-axis) was utilized in the virtual environment after weldment deposition, illustrated in Figure 14, since this is the most reliable continuous processing package due to the limitation of the robotic axis. The first phase of virtual machining was organized in two passes due to the limitation of the cutting depth to 0.5 mm, leaving an additional 0.5 mm of material for finishing. When milling the shapes with ball mills, the angle between the milling axis and the normal position of the surface at the machining point was adjusted to improve tool life and machining quality. For coarse milling operations, the tool angle was set to 30°, i.e., the first phase.

The final stage of virtual machining (phase II) was also staged by rotating the piece from the bottom up, without interruptions, through the entire height of the piece, as shown in Figure 15a. In the second phase of processing, a smaller cutting width (*a_e_*) and milling depth of cut (*a_p_*) were used. After machining the side surfaces, the upper region was machined, for which a rectangular orientation of the tool was retained as per the illustration in Figure 15b, while a one-way machining principle was sustained with the same parameters as in the machining of the side surfaces from the same figure.

Figure 16 shows the finishing operations as phase two on the rough machined blade component. Figure 16a illustrates the side surface from the rotational one-way bottom-up strategy while Figure 16b adopts to the finishing of the top surface in rectangular orientation. A collision may also occur when the cutter approaches the workpiece during operation, so to avoid this, the distance from the workpiece at which the switch between a fast and working motion occurs was set to 10 mm. The theoretical surfacing times calculated were significantly shorter than the machining times due to the higher deposition rate of the material. The first phase of processing each segment took about 60 min. Due to errors in the accuracy of surfacing, the larger addition of material implemented in the first phase of processing was carried out in two passes. This avoided an excessive depth of cut. The planned final treatment lasted 5 h and 40 min due to the smaller milling width set and the requirement to achieve the lowest possible roughness of the treated surface. The total theoretical surfacing time was 53 min, which in the case of selective laser melting (SLM) can be approximated at 21.25 h for a 400 g workpiece based on our previous evaluation (at a maximum of ~6 g/min for a flat surface only).

## 5. Experimental Hybrid Processing of AlSi5 Work Piece

According to the defined plan of hybrid production in a virtual environment and the selected WAAM and cutting parameters, the verification of the process was performed on fabricated prototypes. During production, the geometric properties of the welded structure (the thickness and width of the layers) and the mass of the workpiece before and after the deposition of each segment were also monitored. The additional data on the mass of consumed material during deposition were used in the calculation of production costs.

### 5.1. Prototype Fabrication

The prototype turbine blade was made in eight recurring stages by combining the robotized processes of wire arc welding and machining by milling. For the adequate deposition of the alloy, the welding parameters were selected based on experiments performed on flat walls, which are defined in Table 2 (virtually for sample B5) and Table 3 (experimentally).

The processing of the turbine blade prototype was performed in two phases. The parameters of the first phase were used for milling after the deposition of an individual product segment, while the second phase, i.e., the finishing run, was performed for the entire piece without interruption, and with a smaller width and depth of milling in the concurrent phase. The two-phase milling parameters are shown in Table 4.

The individual stages of production are shown below, with each Figure 17a,c,e,g demonstrating the condition after alloy deposition while Figure 17b,d,f,h represent the second optimized build (Table 4) of the blade segment after subsequent milling on the 20 mm deposit. Figure 17a indicates the initial 20 mm of the AlSi5 alloy deposition according to the welding parameters suggested in Table 3, with coarse machining applied to the deposited segment according to Table 4. The additional 20 mm of deposit was developed as in Figure 17c, which was trimmed to a 40 mm range by the milling process shown in Figure 17d. Likewise, Figure 17e shows direct energy deposition in the range of 40–60 mm, which was machined to specifications in Figure 17f. The structure was extended to 80 mm in Figure 17g, indicating an additional deposit of 20 mm, which was machined until the follow-up stage shown in Figure 17h.

Similarly, the intermediate (fifth) fabrication stage commenced following the 80–100 mm weld deposition shown in Figure 17a, in which the workpiece was trimmed by machining—shown in the right-side image of Figure 17b—to a height of 100 mm. The sixth stage increased the wall height to 120 mm (Figure 17c), which was later milled to specifications as shown in Figure 17d. The seventh stage extended the deposit from 120 mm to 140 mm as in Figure 17e, which later was reduced by milling to an appropriate height, as shown in Figure 17f. 

The final deposition stage (eighth) resulted in a wall height increase from 140 mm to 153 mm as in Figure 17g, which was coarsely milled yet again in the follow-up as in Figure 17h. The fluctuation in the average layer height of the different segments is approximately 0.2 mm. The height of the layers over the entire piece was on average 1.73 mm, which is slightly less than the modelled height during the designing of the welding parameters (1.8 mm). The average wall thickness did not exceed 6 mm.

Table 5 presents the production data on the measured section heights, the average layer heights and thicknesses, and the mass of the welded material in the prototype stage. Following deposition and phase 1, the coarse machining process shown in Table 5 was performed until the total workpiece height of 153 mm was achieved. Figure 18c shows a cross-sectional slice of 11 layers formed in 8 weld deposits.

The final phase 2 milling operation on the 153 mm tall turbine blade segment was performed as a finishing operation, which is illustrated in Figure 19 for (a) the side and (b) the top view. The final tolerances were later compared by 3D scanning of the workpiece from the side and top. These results fit well with the CAM virtual simulations. Figure 19c shows the cross-sectional view of the macrostructure detailing good nominal control over the total 11 interlayers deposition which was further refined by two-phases of milling operations on each deposited layer.

### 5.2. Reverse Engineering Approach for Dimensional Deviations Quantification

In the SprutCAM program, a model for welding and machining was defined in a virtual environment, and based on the optimized parameters, the prototype’s fabrication was undertaken. The surface tolerance difference between the CAD model and the real workpiece was evaluated with surface scanning using the Renishaw Cyclone 2 scanning device to obtain dimensionally unknown 2D-profiles and 3D-surfaces by touch probe rastering, which were then transferred to generate a Solidworks CAD model. Here, a reverse engineering approach is utilized for the quality assessment of the prototype by 3D scanning and forming a 3D CAD model to compare the tolerances between the CAD model and the real workpiece, as shown in Figure 20.

By measuring the geometric accuracy of the prototype compared to the Solidworks CAD model under the reverse engineering mode, it was assessed that the deviation in robotic milling processing was within the expected range. Based on the measurements of positional accuracy performed in the processing of prototypes from previous studies, we expected deviations in the range of ~0.8 mm. The average of the measured deviations on the test piece between the CAD/CAM virtual simulation and the experimental procedure was approximated to −0.76 mm and a standard deviation of 0.22 mm. The difference between the maximum and minimum error was 1.11 mm, as shown in the deviation plot in Figure 21. The workpiece was located at 1700 mm from the base; thus, the static stiffness characteristics were appropriate for machining, while providing sufficient maneuvering space for the robot in order to move the milling head and prevent collisions. The cutting force depends on the depth and width of the chip; hence, when utilizing *a_p_* = 0.5 mm and *a_e_* = 0.3 mm, the stiffness of the robot must not significantly affect the machining accuracy with the set cutting parameters [44]. Contrary to expectations, the measurements in Figure 20 indicate excessive material removal due to the rigidity of the robotic arm and the forces that occur in a dynamic process, e.g., milling; these deviations correspond to over-processing [7]. The vibrations of the robotic arm occur during the feed movement of the cutter at 600 mm/min due to the rapid and sharp turns of the individual axes of the robot that reflected on the surface condition. The high surface roughness resulting from dynamic loads could be improved by reducing the cutting feed rate and by using machining strategies in which the required axis rotations are smaller.

### 5.3. Protype Surface Roughness before and after Machining

During WAAM deposition, corrugation (grooving) of the surface occurs as in Figure 22a under the current parameters specified in Table 3, which need to be eliminated by the subsequent milling process. The order of corrugation size was determined by the surface profile image of the weldments in Figure 22. Here, Figure 22c shows the surface topographical profile in the direction of the walls’ construction after the fifth phase (100 mm), where the corrugation of the welded surface is clearly visible. The measured surface grooving is 250–370 μm, which is marked along the red line in Figure 22b, and which is possibly due to the larger height of the deposited layers. At greater heights, in order to reduce these surface corrugations, higher welding currents and WAAM torch input energies can be vital for achieving superior remelting, thus remediating weldments by achieving a lower height for each layer [6,42].

After phase 2 of milling, the image of the surface profile measurement positioned transversely to the direction of the milling cutter (parallel to deposition) can be seen in Figure 23, which implies that the subsequent machining procedure eliminates the corrugation during surface finishing. However, the corrugations were still present between the layers due to the rounding of the ball-milling cutter and the milling width. The corrugations positioned transversely to the machining direction range from 10–20 μm, which are substantially smaller than those observed in Figure 19 prior to robotic comminution.

Correspondingly, Figure 24 demonstrates the measurement of the machined surface profile in the direction of the tool feed movement. The corrugation of the surface was approximated at about 50 μm, which could be inferred due to vibrations generated during the movement of the robot.

The relatively high roughness shown in Table 6 of the final stage of milling the prototype can be attributed to the poor rigidity of the robotic arm, which is particularly reflected in the processing of ductile metals and alloys. The machining procedure was performed rotationally, with the tool turning around the workpiece; however, in certain positions, the rapid rotation of the fourth, fifth, and sixth axes of the robot is essential, which affects the shaking of the cutter and, consequently, the quality of the treated surface. The probable solution recommended for future research suggests using a revolving table for such rotary processing [4].

According to the comparison of the average surface height (*S_a_*), the roughness improved by approximately 55% with the mechanical comminution. The maximum height of the bulges/protrusions (*S_P_*) and the maximum depth of the bulges/depressions (*S_V_*) after machining also show about 50% lower values were obtained than after the weldment deposition; however, these values are still high. The maximum height of the protrusions was 82 μm, and the maximum depth of the depressions was approximately 93 μm. The condition of the surface could be improved by reducing the speed of milling (extending the processing time), but there is a definite compromise between quality and productivity [46]. Another possibility for improvement constitutes a change in the machining strategy by approaching the movement reduction in the last three robotic axes that reach the highest rotation speeds during machining.

### 5.4. Microstructural Analysis of the Deposited Alloy

The mechanical properties of the material can also be predicted by a microstructural analysis. During welding, a large local energy input is required, which melts the filler material and heats it in the case of the AlSi5 aluminum alloy above 630 °C [47]. To ensure the appropriate weld geometry during WAAM, it is necessary to maintain a certain inter-welding temperature, which, in this case, was lower than 120 °C. In case of large temperature changes, an inhomogeneous (dendritic) crystal structure can be expected, which affects mechanical properties—tending to be anisotropic [36]. To examine the material microstructure, the sample wall was wire arc additive-manufactured with the deposition parameters from the test piece in Table 3, having dimensions of 5.5 × 20 mm and including 11 layers. The sample cross-section was prepared by cutting the wall transversely to the welding direction, i.e., in the direction of wall growth, as shown in Figure 25.

The porosity defects seen in the walls’ cross-sectional magnification images are round in shape (Figure 25a), thus implying that these are gas bubbles that remained trapped in the melt-pool during cooling. The gas bubbles vary in size by up to 95 µm in diameter and are homogeneously distributed throughout the wall. Due to the rapid heterogeneous cooling, these gas bubbles remain trapped during dendritic crystal structure formation, suggesting the improper venting of gases from the melt, which is apparent from Figure 25b. The formation of gas bubbles can be caused by the contamination of the filler material with moisture, uncleaned base material, or the moisture in the shielding gas mixture [6,7]. In our case, the cause of the porosity was moisture trapped in the filler material since the welding trials were performed using the same filler within several weeks. During this time, additional care was not implemented to properly store the filler wire when it was not being used and the filler was left in the welding power source the whole time, without any heat treatment to remove the moisture. In order to mitigate the porosity, a higher-purity argon shielding gas should be used, the filler material should be cleaned, and moisture should be removed, and the linear energy input during welding should be increased to increase the melt-wettability on the surface [1,2]. This would result in a more flattened and wider shape of the individual layers due to higher melting, but with less rapid cooling, the gases would be released from the melt more easily [18]. Otherwise, the CMT-PADV mode can be applied to minimize porosity and moisture effects by controlling the WAAM deposition factors (the oxide-cleaning effect) [32,33]. The prototype material can be homogenized to reduce the dendritic structure to a more finely distributed high-strength acicular silicon-rich secondary phase along with smaller Mg-containing precipitates in the AlSi matrix. Gu et al. [36] confirmed that CMT-WAAM deposition led to hierarchically dispersed dendrites as well as equiaxed grains, with a scarce population of columnar grains, which was also observed in the present study. Such a dendritic microstructure without heat treatment for an equilibrium microstructure results in anisotropic mechanical properties of the AlCuMg alloy. The larger Si-rich precipitates may promote ductility reduction and should be refined by thermal treatments in order to realize the high-strength characteristics of the AlSi5 alloy [4,5]. Zhang et al. [35] suggested that micro pores in the interlayer pore region following the CMT WAAM of Al-alloys lead to anisotropy in the mechanical properties.

### 5.5. Tensile Test Measurements

Tensile strength measurements were taken to determine the mechanical properties of the WAAM structure made of the AlSi5 aluminum alloy. To produce test specimens, the same weld-surfacing parameters as those used for the construction of a test piece were employed, namely, the use of a welded flat wall and an average thickness of 5.5 mm following the DIN 50125: 2009-07 standard. The test specimens were thinned by milling them to the prescribed sample thickness of 3 mm, and the tubes were then cut using an abrasive water jet. The tensile measurements were performed on a total of 12 samples, with five samples having a longitudinal orientation of layers (an angle of 0° with respect to the base surface) and seven samples with a transverse orientation of welding layers (an angle of 90° with respect to the base).

The results of the tensile tests showed small differences in tensile strength between the specimens for each orientation and we also compared a small difference in strength between the transverse and longitudinal orientation, which, on average, was only 0.5 MPa higher in the longitudinal specimens, as organized in Table 7. This explains the rather isotropic nature of the fabricated workpiece, even though a dendritic microstructure was predominant in the non-annealed state. Horgar et al. [20] suggested that the isotropic microstructure enabled yield strength (YS), Ultimate Tensile Strength (UTS), and hardness values superior to the commercial alloys from wire + arc additive fabrication. From the shape of the fracture surfaces illustrated in Figure 26a, it was observed that in the case of the longitudinal patterns, the fracture surface is flat, and runs at an angle of approximately 45° with respect to the direction of loading. Whereas, in the transverse orientation of the samples from Figure 26b, the fracture surface is more varied, as it passes through places where the porosity is higher, and the fracture surface increases, with the loading force distributed to the larger area. Shown in Figure 27, the yield strength was 6.6 MPa higher at the 0° angle (b) than at the transverse orientation in (a); moreover, the elongations in the longitudinal direction were smaller, indicating slightly better toughness of the material in the welding direction. Quantitative evaluations of the stress–strain plots in both the transverse and longitudinal directions from Figure 27a,b are presented in Table 7.

The reason for the results obtained can be explained by observing the sites on the samples where the breakage occurred. In the transverse specimens, failures occurred at different locations in the specimens’ observed area, while in the longitudinal orientation of the samples’, fractures were observed along the same region on the specimen. When surfacing the walls for tensile tests, due to the different degrees of heat dissipation, a slight ripple appeared in the first layers, which was also noticeable in the upper layers at the same distance from the edge [18,19]. At this point, the localization of mechanically inferior material properties and thus lower tensile strength may have resulted. Zhang et al. [35] suggested that generic WAAM led to the formation of dendritic and columnar grains with an interfacial microporosity that tends to retain anisotropic mechanical properties in the fabricated Al-6Mg alloy. However, the closeness of the quantitative strength values in the transverse and longitudinal directions from our experimental study confirms the rather isotropic behavior of the AlSi5 alloy even with the formation of dendrites. One reason for this isotropic behavior can be attributed to the dissolution of Si-rich precipitates in the matrix that causes grain refinement of the alloy’s microstructure. The mechanical properties of the commercial AlSi5 alloy (4043–H18) are: *YS*_0.2_ = 40 MPa, *UTS* = 120 MPa, % Elongation ~8–18%, and Vickers’s hardness of *H_V_* = 87. Comparing the mechanical properties of WAAM AlSi5 from Table 7 with the commercial alloy demonstrates that the YS and UTS for both directions are at least 33% higher and possess ~20% better ductility through hybrid processing. These results make a strong claim for the utilization of the hybrid-manufacturing route over the conventional processing method [9] for this commercial welding material [34]. Follow-up thermal treatments should be expected to enhance the microstructure-linked mechanical properties as more refined secondary precipitates form and dendrites resolve to equiaxed grains [26,27].

### 5.6. Combined Processing Time and Cost Analysis

By calculating the cost of constructing the planned piece, the combined surfacing and milling process can be compared with other established addition processes from industrial expansion and feasibility perspectives. The costs of additional material and shielding gas are included in the calculation, as well as the surfacing costs and mechanical machining costs incurred by milling. The value of the surfacing base plate was neglected due to the small amount.

#### 5.6.1. Cost of Additional Material

The mass of the consumed filler material was obtained by weighing the workpiece before and after surfacing. The sum of the differences in weight represents the total consumable additional material. Its cost is calculated according to Equation (1).
(1)$$S_{dm}=m_{dm}\cdot c_{dm}$$
Here: *S_dm_* = cost of deposited material (EUR);*m_dm_* = mass of filler material (g);*c_dm_* = price of deposition material (EUR/kg).

#### 5.6.2. Shielding Gas Costs

When calculating the cost of shielding gas according to Equation (2), we used the set value of gas flow, total welding time, and the price of consumed argon.
(2)$$S_{pl}=\dot{V_{pl}}\cdot t_{v}^{N}\cdot c_{pl}$$
where: *S_pl_* = cost of shielding gas (EUR);${\dot{V}}_{pl\;}$ = shielding gas flow rate (L/min);$t_{v}^{N}\;$ = deposition time (h);*c_pl_* = price per liter of expended gas (EUR/L).

#### 5.6.3. Weldment Deposition Costs

When calculating labor costs, the time for the indirect programming of the surfacing and preparation of the NC code, the surfacing times, and cooling pauses during welding to reach a certain inter-welding temperature, and the preparatory end time, which contains the time to prepare the post, calibration, and clamping at the machining site during the manufacture of individual pieces of the main workpiece, were collectively considered. The depreciation of the robotic welding cell and energy consumption are considered in the price of the machine clock. The surfacing cost was calculated according to Equation (3).
(3)$$S_{n}=c_{h}^{N}\cdot \left(t_{n}^{N}+t_{k}^{N}+t_{pz}^{N}+t_{v}^{N}+t_{o}^{N}\right)$$
Here: *S_n_* = Deposition cost with robot (EUR);$c_{h}^{N}$*=* Price robotic welding cell working per hour (EUR/h);$t_{n}^{N}$ = Deposition planning time (h);$t_{nc}^{N}$ = NC code preparation time (h);$t_{pz}^{N}$ = Preparation-finishing time (h);$t_{v}^{N}$ = Deposition time (h);$t_{o}^{N}$ = Cooling time during deposition (h).

#### 5.6.4. The Cost of Mechanical Processing

The calculation of the machining time was considered when calculating the machining costs, total processing time, and preparatory closing time wherein the time to perform calibrations is included. The preparation of the workplace and the tuning of the workpiece during the individual stages of processing are considered. Machine depreciation and energy consumption are included in the hourly rate of working with the robot.
(4)$$S_{f}=c_{h}^{F}\cdot \left(t_{n}^{F}+t_{pz}^{F}+t_{o}^{F}\right)$$
where: *S_f_* = Cost of milling with the robot (EUR);$c_{h}^{F}$ = Cost per hour of working with milling robot (EUR/h);$t_{n}^{F}$ = Processing scheduling time (h);$t_{pz}^{F}$ = Preparation-finishing time (h);$t_{o}^{F}$ = Total processing time (h).

#### 5.6.5. Total Costs of Combined Production of a Test Piece

The total cost of manufacturing a test piece with a combined process of the robotic surfacing and milling of aluminum is determined by the sum of cost centers, which is calculated by the following Equation (5) and summarized in Table 8.
(5)$$S=S_{dm}+S_{pl}+S_{n}+S_{f}$$

The test samples were made from Table 2’s (B-5) parameters, with the first one as the initial build to validate the CAM environment (virtual processing), whereas the fabricated AlSi5 alloy blade in Figure 17, Figure 18, Figure 19, Figure 20, Figure 21, Figure 22, Figure 23 and Figure 24 correspond to the optimized build. The cost of fabrication for these two cases with B-5 parameters is detailed in Table 8, which indicates that the optimal build is more than two times cheaper to produce due to the calculated time and resources disbursed during the weldment deposition and subsequent milling processes. The times for each of these operations were recorded as presented in Figure 28 for alloy deposition and Figure 29 for the milling process.

It can be seen both in Figure 28 and Figure 29 that the robotic path planning in the CAM environment requires 3.3–4 h during the first build, which was subsequently reduced to nearly half with the optimization of the B-5 parameters (Table 2). Evidently, by comparing Figure 28 with Figure 29 with respect to the cost data in Table 8, it can be inferred that the largest share of cost and time was taken by the milling operations, whereas the CMT MIG material deposition remains a rapid and cost-efficient process. The surfacing of weldments, as already described, is very critical for achieving enough rigidity and mechanical strength of the optimized blade; therefore, the use of robotic milling is strongly advocated for such an intricate geometry. The total time taken to construct the prototype was approximately 35.1 h (to fully complete the job) as derived from the CAPP conceptualization, with the CAM virtual simulation and eventually part production, is shown in Figure 19. The breakdown of energy consumption comprises (a) the welding robot = 1.65 kW and welding system (effective consumption) = 3.75 kW, and (b) robotic milling at 1.65 kW for the robot, the spindle requiring 2 kW, and the cooling system operating at 2.5 kW. In total, the energy consumption for the initial prototype accounts for 5.4 kW for the welding deposition system and 6.15 kW per hour for the milling unit. The resultant data correspond to a total hybrid fabrication consuming 11.55 kW of energy per hour, which surges to 405.41 kW during the whole fabrication timeline of 35.1 h. However, with the control in place, the optimized build took an estimated 14.2 h to complete the same part. Thus, the total energy consumption was successfully and sustainably reduced from 405.41 kW to merely 164 kW for the fabrication of the optimized build within 14.2 h. 

A processing time analysis of the hybrid fabrication was also conducted for the Selective Laser Melting (SLM) of the AlSi5 alloy (non-published; raw material with same composition). It was deduced that the total SLM fabrication took approximately 21.25 h in comparison to the 14.2 h of hybrid fabrication based on the B-5 parameters from Table 2. The SLM method required approx. 4 kg of AlSI5 alloy, while actual powder consumption is 324 g for the workpiece up to 160 mm height, and an additional 28 g for the supporting base. On average, the energy consumption of running the SLM machine is 8.5 kWh. Thus, for the total fabrication time of 21.25 h, the consumed power can be expected to be around 180 kWh.

The cost of the AlSi5 filler material in CMT MIG deposition was the same—around EUR 6—based on less than 500 g of material consumed per fabrication, whereas a minimum of 4 kg of Al powder needed to be added to the SLM system for processing a 325 g workpiece on a 28 g base. Since only Ar shielding gas was consumed in the processing during hybrid fabrication, the costs are relatively lower than SLM, which has to account for chamber purge, zone cleaning, and continuous flow during the part’s manufacture. Moreover, additional system preparations and, later, cutting from the base resulted in EUR 220 spent in this section. Comparatively, in hybrid fabrication, the initial part consumed EUR 35 of shielding gas over a period of weldment deposition of 15.1 h, and this cost was reduced to only EUR 14 with processing optimization (5.8 h only). Extensive planning with CAPP + CAM and robot calibration initially incurred EUR 273 due to energy consumption, whereas MIG deposition for the initial build used up ~EUR 218, which was further lowered to only EUR 148 as the hybrid deposition and processing were shortened. Similarly, the CAM setup and calibration of the milling robot incurred ~EUR 313 over the course of 8 h, and the total robotic milling operation initially costed EUR 977. Later, with the reduction in processing times, the robotic CAM preparation/calibration and milling resulted in approximately EUR 140 and EUR 232 spent, respectively. 

The largest share of the manufacturing cost is the time of milling, in which most machine hours were spent. By optimizing the welding parameters and improving the surfacing accuracy, the amount of material that needs to be removed by milling could be reduced [3,4]. In this way, the piece could be processed in just one pass, which would reduce processing costs by approximately EUR 1000. Combined production for smaller batch sizes is less accurate compared to the data obtained, but system upgrades would significantly improve it. The production took place with two robots with different clamping tables, which made it necessary to reposition the workpiece between the individual stages of processing; thus, more time was expended than optimally intended.

As a comparative cost evaluation versus hybrid fabrication, the SLM experimental data (non-published) reveal the total price of a single AlSi5 blade at EUR 1725. As explained before, the additional raw material cost was EUR 44.7; the preparation cost for Ar shielding, cleaning, and the cutting of the base was EUR 220; and the printing hours incurred a charge of EUR 1487.5 for the total duration of 21.25 h. For additional printing with the same base, the cost of parts reduced to ~EUR 898.5/pc, whereas for the fabrication of four blades simultaneously, SLM offers a cost value of EUR 511/pc. To compensate the cooling time, robotic milling can be adjusted for additional parts. Thus, multiple parts with a high degree of design freedom fabricated by the hybrid route offer a more reasonable and pragmatic yet sustainable solution compared to smaller batch sizes manufacturing with SLM. The highest share of time expense in hybrid processing was attributed to the welding robot’s set-up and cooling time instead of the deposition itself. Thus, multiple parts with the same product model can be fabricated under the optimum conditions simultaneously to elude time wastage from waiting during the cooldown stage. Hence, the material deposition is convened in parallel for the second part, while the first one receives milling augmentation, and so on. In analogy to the previous work on AlSi10Mg alloys [5], the SLM is limited by parts’ dimensions, and single piece fabrication is both energy- and cost-intensive, whereby the dimensional tolerance is usually in range of ±300–3000 µm with an *R_a_* < 5 μm based on the deposition parameters [7]. The hybrid processing returned a dimensional accuracy of ±760 µm with an *R_a_* ranging up to 14.3 μm and delivering a product volume of 55 m^3^, which is several orders of magnitude larger than SLM. Nonetheless, the sustainability of manufacturing multiple parts with SLM and hybrid fabrication is remarkably competitive; however, SLM, in terms of energy consumption for a single piece, is considerably more expensive than the WAAM route suggested in this study.

With the improvement of the welding robot’s postprocessor, manual code preparation would no longer be necessary, so this time can be subtracted. Due to the treatment at one clamping point, the preparation and finishing times would be reduced, and the installation of a cooling system would help reduce the surfacing time, which would allow the cooling times to reach at least half-temperature. The largest share is the pause time during welding to ensure the appropriate inter-welding temperature and represents about 80% of the welding time. Costs could be reduced here as well by employing an appropriate cooling system at a constant temperature or producing more workpieces at the same time. Convection cooling is not sufficiently intense in air, so heat accumulates in the product. An easy possible solution would be to mount the weldment on a water-cooled base. The costs of additional material (which applies to the AlSi5 aluminum alloy) and shielding gas are negligible in terms of the design and operation costs of the robotic cell [4,8]. Experience in planning machining paths would help to accelerate the preparation of machining, and the preparation and finishing times could be reduced by working in one clamping point. By optimizing the welding parameters and through the more accurate application of the material, the processing could be performed in just one pass [6]. The processing time and thus costs would be significantly reduced. The reason for the greater roughness lies in the specifications of the robotic processing by milling, which is especially evident in the processing of metallic materials [8,13]. Prior to mechanical treatment, the welded wall’s surface is corrugated and incomparable with other processes. Laser remelting and dust loading take place in thinner layers, which enables a relatively low roughness even up to *R_a_* = 2–5 μm [5,18]. However, better CMT-welding techniques offer the promise of finer droplet deposition on the weldment and potentiate reduced machining runs, eventually increasing productivity [3,4].

Regarding the surfacing of metals, due to temperature differences, the base material experienced thermally induced deformation, which further complicated the exact positioning of the workpiece [2,12]. With a built-in cooling system for faster heat dissipation, this time could be reduced to 50%. The preparation of the NC code for the surfacing process had to be automated by improving the postprocessor, which would further reduce the preparation time. The cutting process represented the largest share of costs in the production of the test piece. By optimizing the welding parameters, the material could be welded more accurately and precisely, thus enabling machining in only one pass, which would reduce the machining cost at least by two-thirds (2/3). This hybrid WAAM + milling approach becomes much more viable and pragmatic for bulk single piece complex components following these system upgrades and the parametric optimization of the prototype, thus enabling actual production to run sustainably in large batches.

## 6. Conclusions

This study evaluated the hybrid fabrication of an AlSi5 alloy workpiece in the form of a turbine blade. The geometric design of the turbine blade poses significant challenges in terms of the multilayer (11 layers) deposition via a robot, followed by the subsequent milling of the surface. The hypothesis behind the current study related to: (a).The implementation of the indirect programming of robots with six degrees of freedom for machining;(b).Stability analysis of robotic WAAM surfacing on the quality of a workpiece’s structure and construction;(c).The development of a hybrid system (WAAM + milling) in a virtual environment, which must be verified by experimental work on a real product;(d).The cross-deployment of simulated parameters to robotic PLCs, in which hybrid processing is implemented on a prototype workpiece that meets the CAD/CAM design specifications and wherein the resulting material properties should meet the commercial tolerances.

It can be concluded that the hybrid process of robotic WAAM surfacing and milling of an AlSi5 alloy can suitably be applied to larger volume products (free-form) with relaxed tolerances. Based on the optimization of the CAD/CAM parameters during the virtual simulation, the generated part geometries, WAAM deposition parameters, automated tool path, robotic milling, and NC-code generation demonstrate the efficiency of self-autonomous AM solutions that can fabricate complex functional workpieces from the product models (DFAM/CAD). The results regarding 3D dimensional accuracy (reverse engineering) showed deviations of approximately −0.76 mm from the planned model; however, the roughness values are still within the simulated range. The precision of processing is affected to the greatest extent by the quality of the tool’s calibration, the positional accuracy of the robot/machine, and the thermal expansion criterion. This can be improved by implementing adequately calibrated tooling and stiffer robotic clamping to enable high-speed operations. Our microstructural analysis showed the occurrence of microporosity from gas bubbles, ranging up to 95 µm in the vicinity of dendritic matrix. Due to the rapid cooling of each welded layer, the gas bubbles could not pass to the melt surface quickly enough to escape from the molten pool. By optimizing the CMT-WAAM parameters and using a cleaner/moisture free filler, the structure of the material could be improved. The anisotropy of the WAAM structures in the transverse and longitudinal directions with respect to the weld-surfacing direction is quite insignificant, and higher toughness in the longitudinal direction was observed. Mechanical testing indicated 33% better tensile strengths than the manufacturer’s specifications in both directions, despite a dendritic microstructure and microporosity. The approximate value of UTS = 160 MPa at an elongation of 25–28% exceeded our initial estimation of mechanical properties. Further refinement in microstructural inhomogeneity via thermal treatments should translate to even better mechanical properties. Lastly, for comparison, an in-house SLM fabrication of a blade workpiece was performed. The hybrid fabrication by robotic WAAM + milling facilitated price competitiveness (in terms of energy, processing time, and total costs) versus SLM. However, the weaknesses are reflected in the calibration of robotic milling unit realizing relatively poor part accuracy and higher surface roughness of the final product. The advantage of the robotic WAAM process concerns its applicability to various cheaper filler materials and larger product volumes. Tightening the tolerances with smart robots operating in energy deposition and machining (a hybrid approach) should compete with even the very best AM solutions in the market.

## Figures and Tables

**Figure 1 materials-15-08631-f001:**
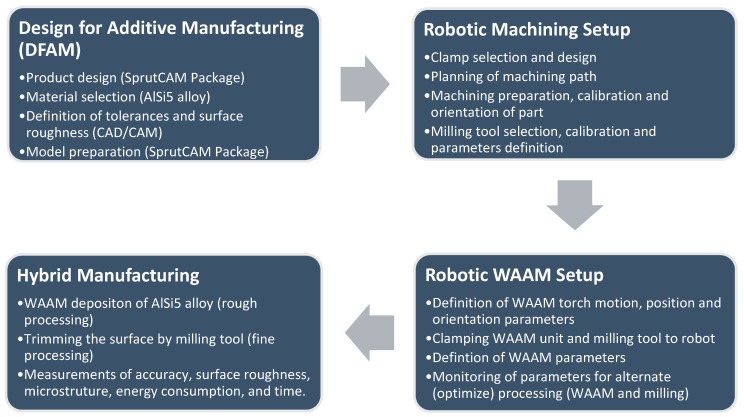
Hybrid-manufacturing process design and scheme applied to produce turbine blade workpiece out of AlSi5 alloy by cooperatively combined robotic WAAM and milling.

**Figure 2 materials-15-08631-f002:**
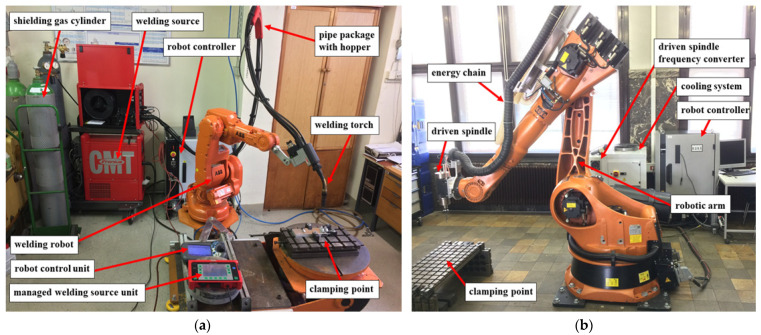
Experimental robotic setups: (**a**) robotic welding cell and (**b**) robotic cell with milling unit attachment.

**Figure 3 materials-15-08631-f003:**
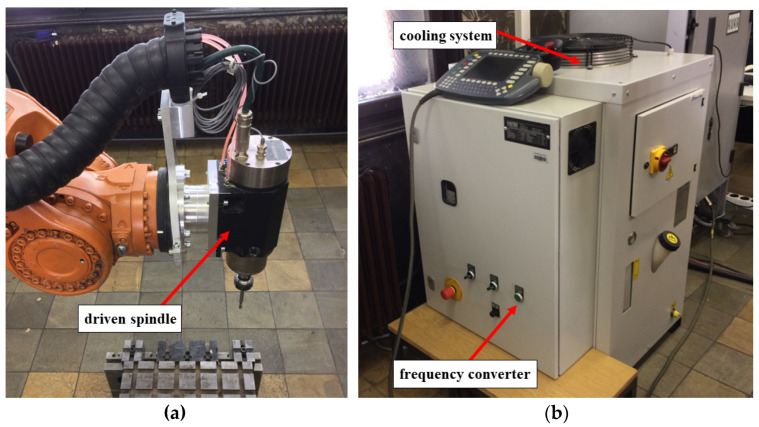
(**a**) Robotic head with electronic milling spindle drive, and (**b**) spindle cooling system with the frequency converter.

**Figure 4 materials-15-08631-f004:**
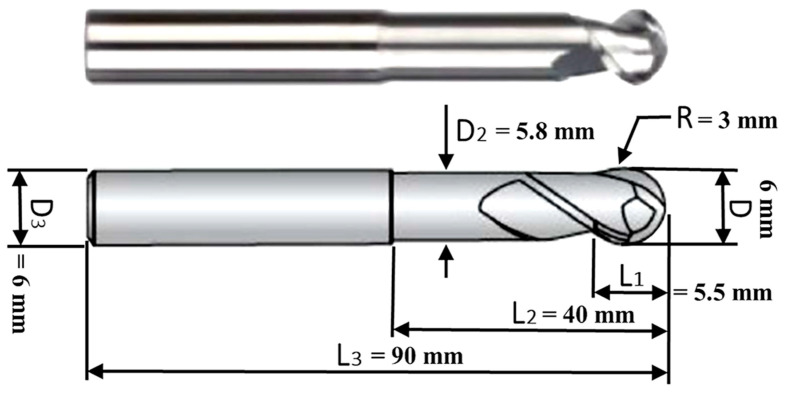
WAB312061 Carbide cutting tool with 6 mm diameter having two cutting edges.

**Figure 5 materials-15-08631-f005:**
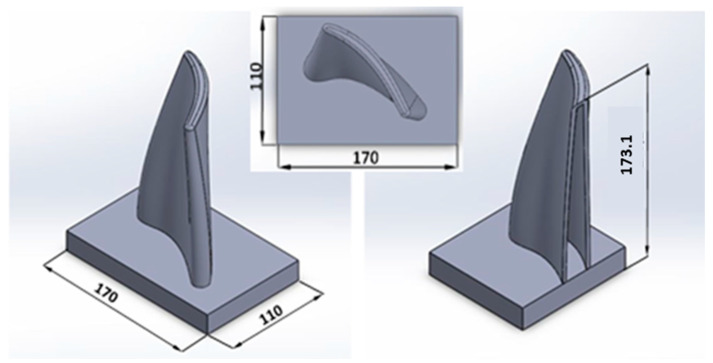
Turbine blade prototype model (dimensions in mm).

**Figure 6 materials-15-08631-f006:**
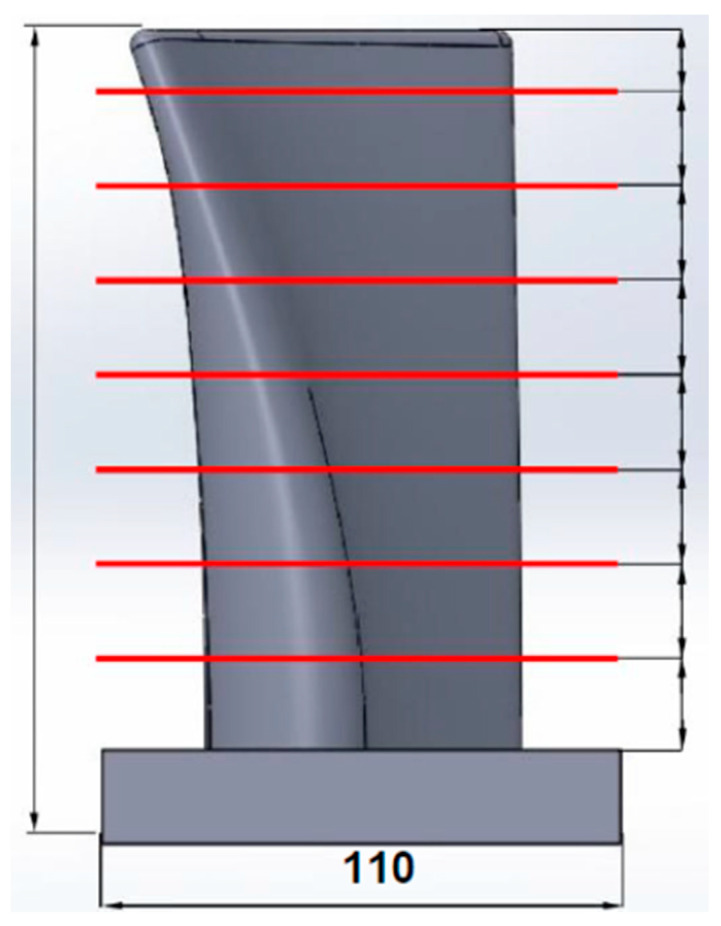
Division of workpiece to eight 20 mm segments, up to length of 153 mm.

**Figure 7 materials-15-08631-f007:**
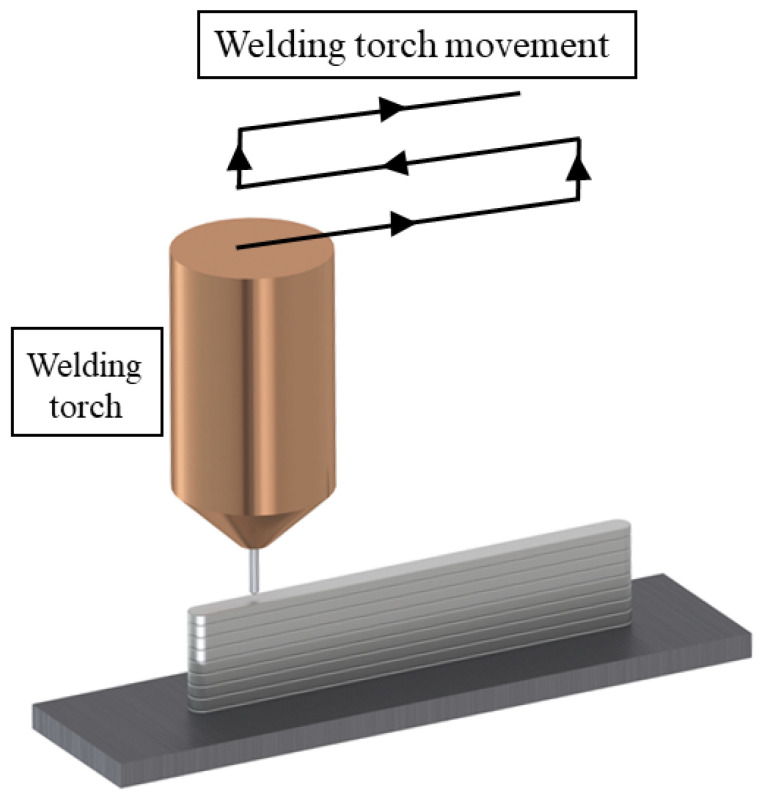
Welding torch movement feed during deposition on flat walls (parallel to base surface).

**Figure 8 materials-15-08631-f008:**
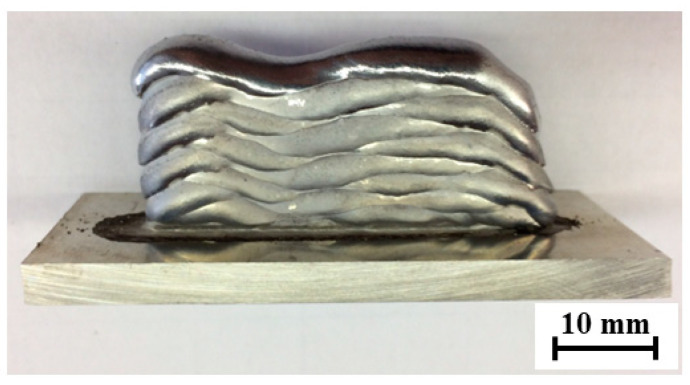
Melting of deposited wall at excessively high inter-layer temperature of >120 °C.

**Figure 9 materials-15-08631-f009:**
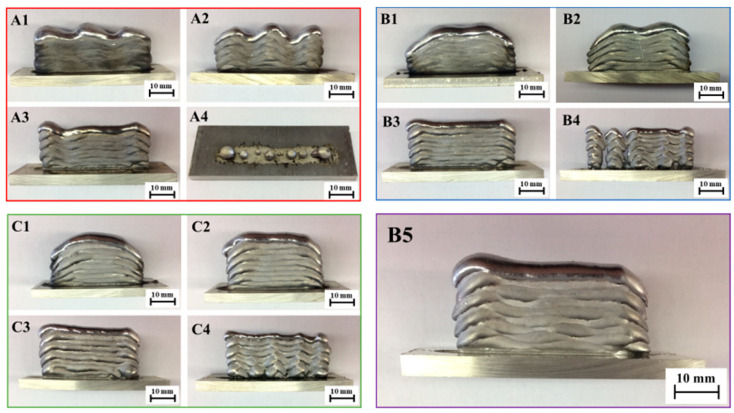
Different WAAM deposition test conditions: A1–A4 at higher welding speed pf 10 mm/s, B1–B5 at intermediate welding speed of 8 mm/s, and C1–C4 at slower welding speed of 6 mm/s.

**Figure 10 materials-15-08631-f010:**
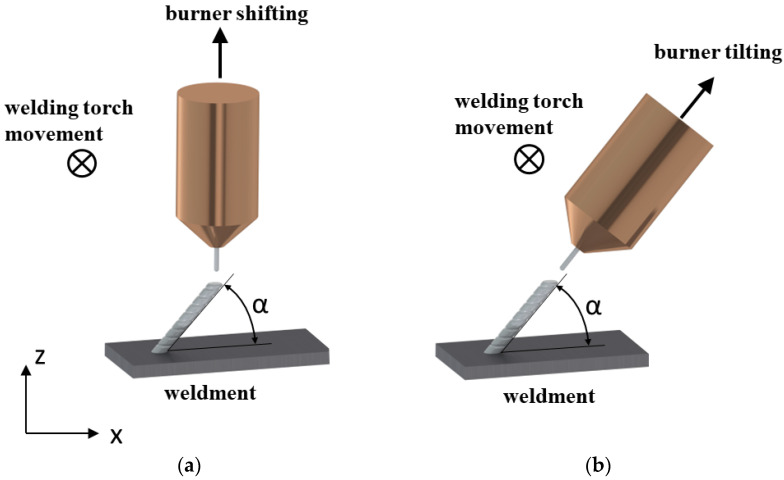
Strategies for surfacing of flat wall at an angled deposition by (**a**) shifting the burner and (**b**) titling the burner.

**Figure 11 materials-15-08631-f011:**
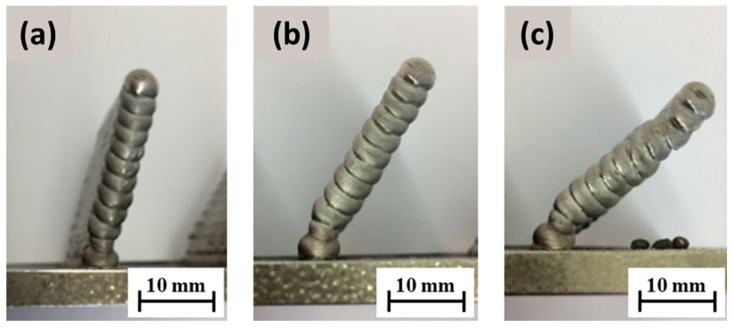
Surface deposition of flat walls at an angle of (**a**) 75°, (**b**) 60°, and (**c**) 45°, according to burner offset/shift.

**Figure 12 materials-15-08631-f012:**
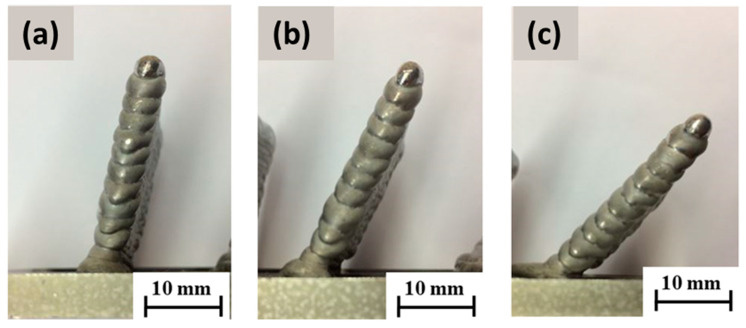
Direct energy surface deposition of flat walls at an angle of (**a**) 75°, (**b**) 60°, and (**c**) 45°, with titled welding torch.

**Figure 13 materials-15-08631-f013:**
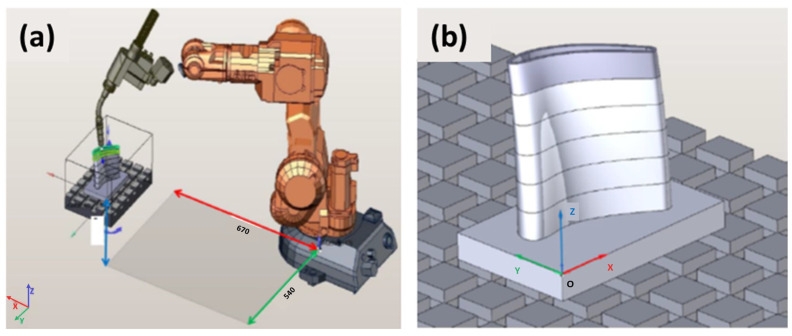
(**a**) Placement of workpiece in work area of welding robot and (**b**) layout of workpiece coordinate system for welding deposition.

**Figure 14 materials-15-08631-f014:**
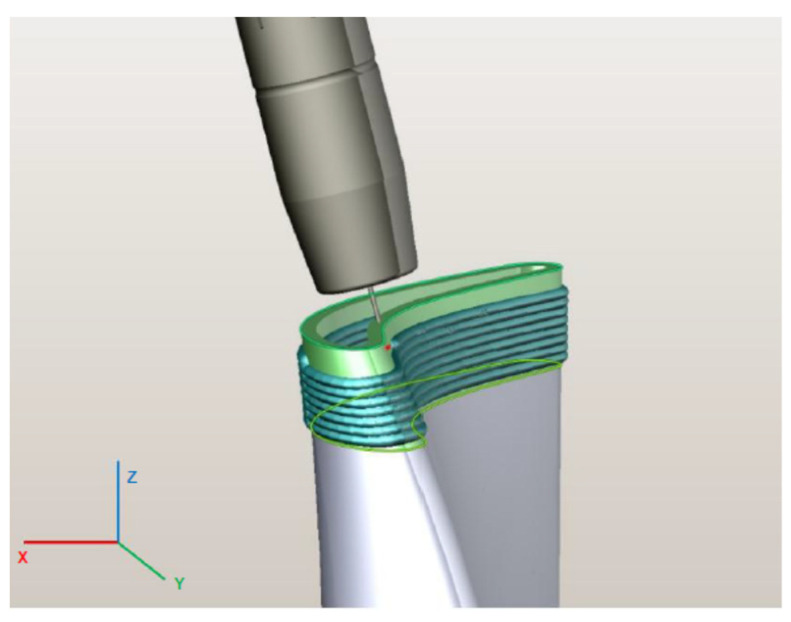
Orientation of welding burner and deposition path (green lines) in the direction of structure construction.

**Figure 15 materials-15-08631-f015:**
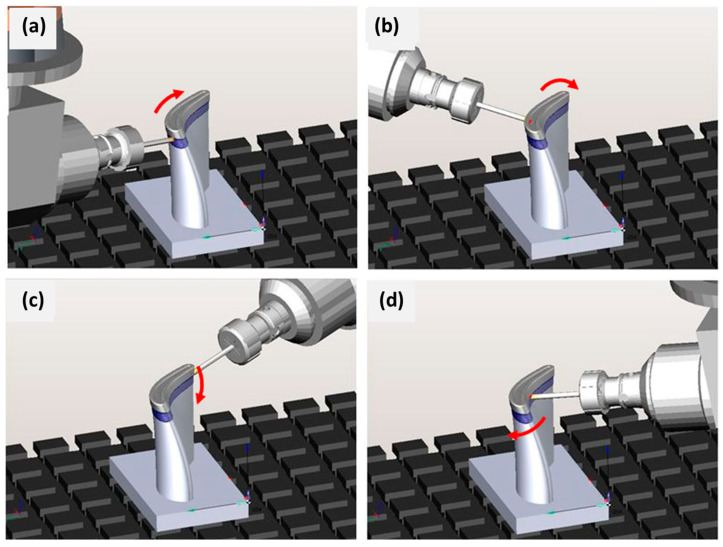
Orientation of tool during machining of the sixth segment, from the (**a**) left, (**b**) rear, (**c**) right, and (**d**) front.

**Figure 16 materials-15-08631-f016:**
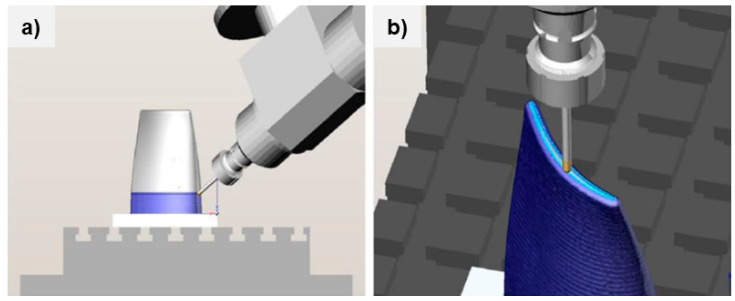
Finishing operations performed on the (**a**) side surface from the rotational one-way bottom-up strategy and (**b**) the top surface in rectangular orientation.

**Figure 17 materials-15-08631-f017:**
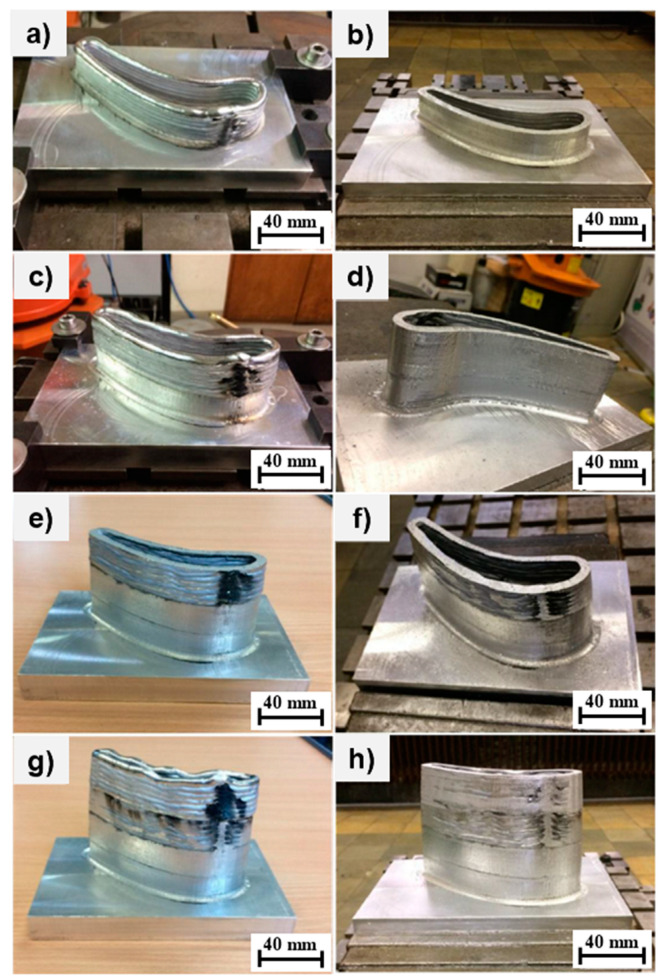
Stages of prototype fabrication by weldment deposition and subsequent milling: (**a**) first stage of deposition from 0–20 mm; (**b**) milling to 20 mm height; (**c**) second stage deposition up to 40 mm followed by (**d**) machining at 40 mm; (**e**) third stage deposition 40–60 mm and (**f**) milling third deposited segment to 60 mm; (**g**) intermediate weldment structure in fourth stage at 60–80 mm and (**h**) follow-up machining to 80 mm.

**Figure 18 materials-15-08631-f018:**
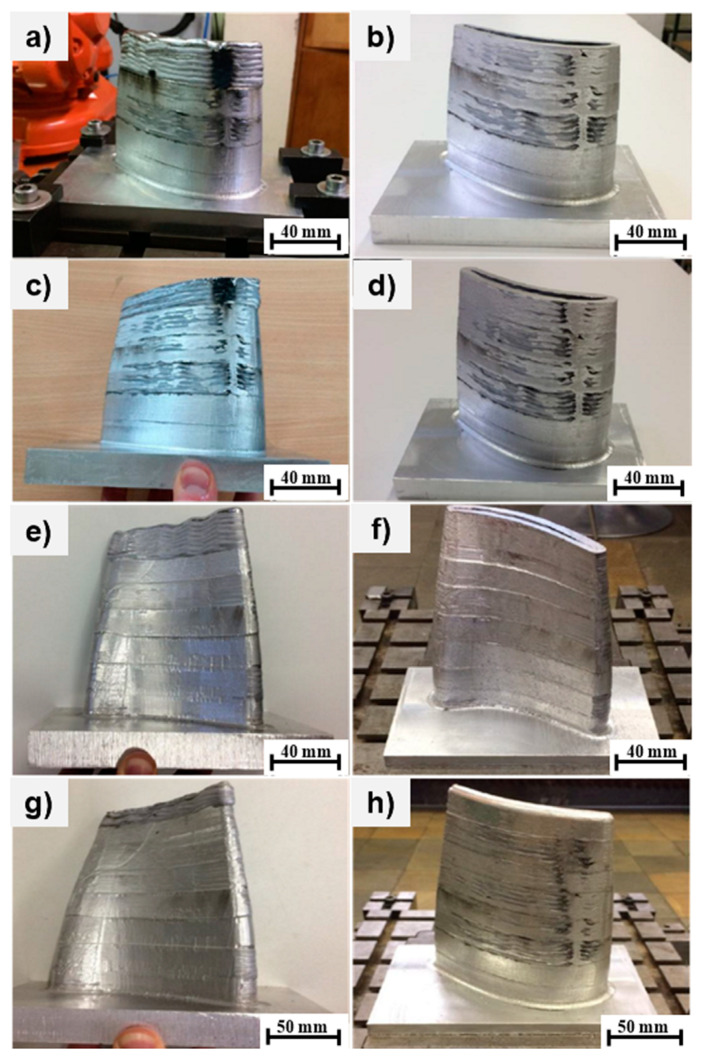
Intermediate to final stage of turbine blade fabrication up to 153 mm in height after (**a**) weld deposition at 80–100 mm and (**b**) follow-up milling to 100 mm; (**c**) the sixth stage from 100–120 mm by alloy deposition; (**d**) machined to a height of 120 mm; (**e**) weldment structure from 120–140 mm and (**f**) subsequent milling of seventh stage to 140 mm; (**g**) lastly, the final deposit in eight stages from 140 mm to 153 mm, which was milled to specifications in (**h**).

**Figure 19 materials-15-08631-f019:**
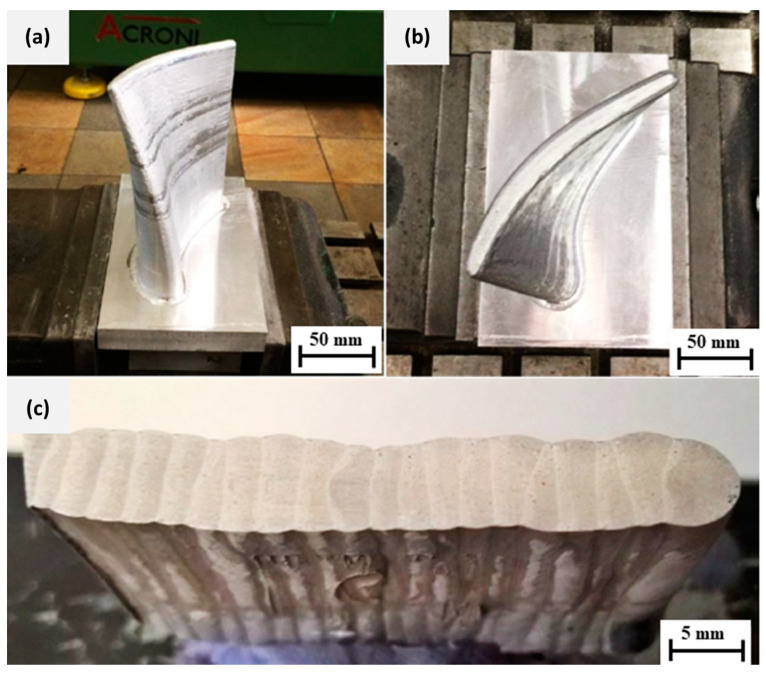
After the final phase 2 machining of the optimally built workpiece: (**a**) side view and (**b**) top view of 8 depositions and 11 layers in total from the (**c**) cross-section slice image.

**Figure 20 materials-15-08631-f020:**
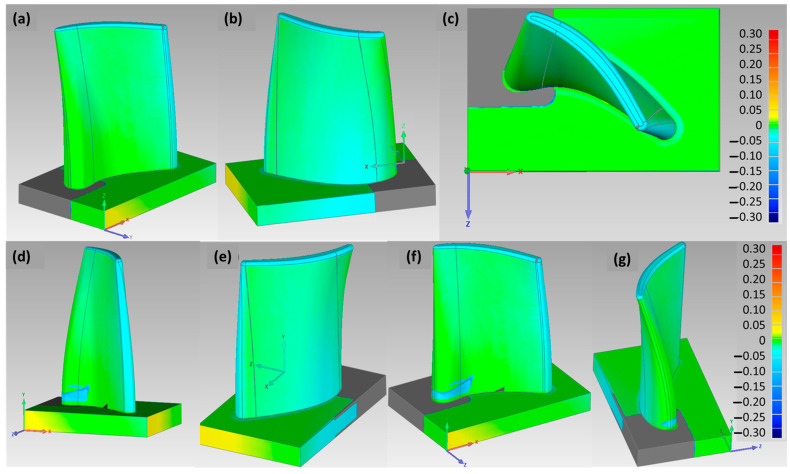
CAD model developed from 3D-scanned prototype showing: (**a**) front view, (**b**) rear view, (**c**) top view, (**d**) side view 1, (**e**) side view 2, (**f**) frontal side view, and (**g**) cross-side view.

**Figure 21 materials-15-08631-f021:**
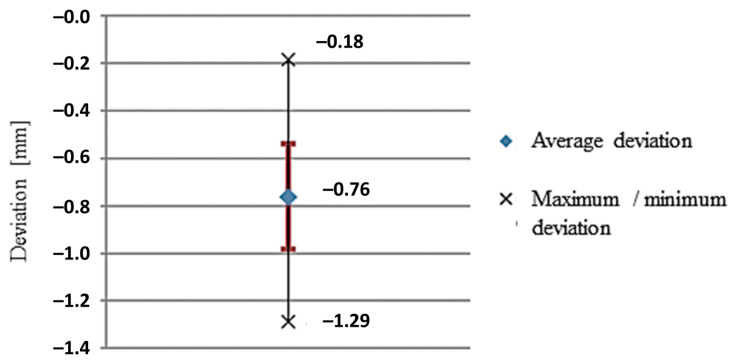
Measured deviation of prototype from CAD model shown with average value, standard deviation, and the minima/maxima values.

**Figure 22 materials-15-08631-f022:**
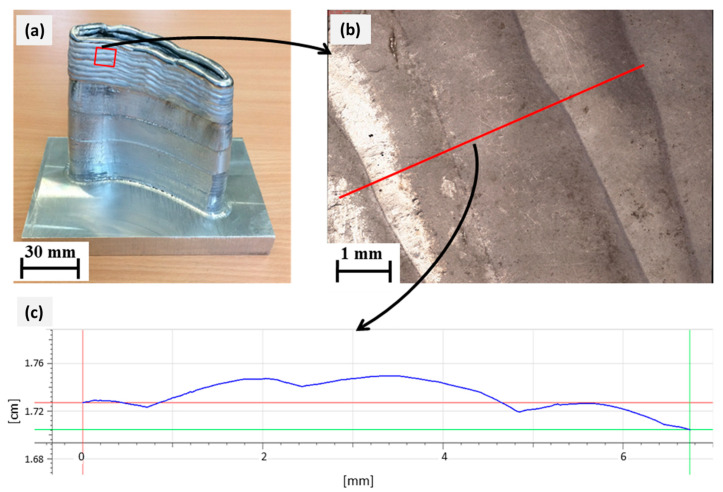
Measurement of the weldment surface at (**a**) 100 mm height in fifth stage of deposition, (**b**) higher-magnification optical image of corrugation between welding layers between 80–100 mm, and (**c**) surface topographical profile.

**Figure 23 materials-15-08631-f023:**
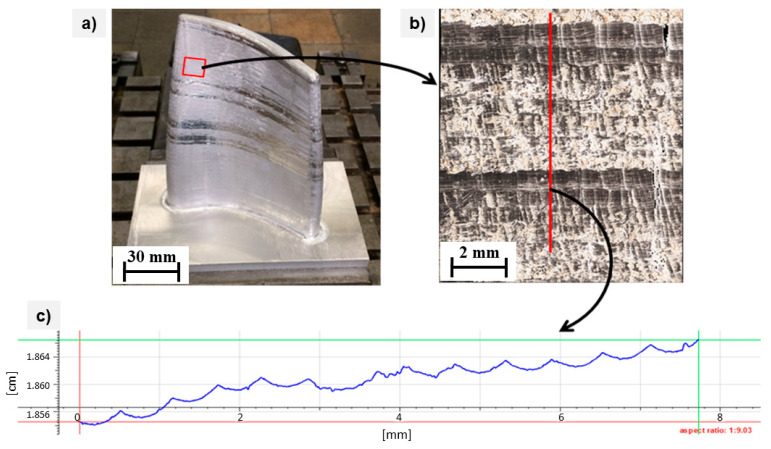
Surface profile analysis in (**a**) finished prototype with second phase milling, (**b**) higher magnification of milled surface, and (**c**) topographical profile measurement along the traverse direction to the milling cutter.

**Figure 24 materials-15-08631-f024:**
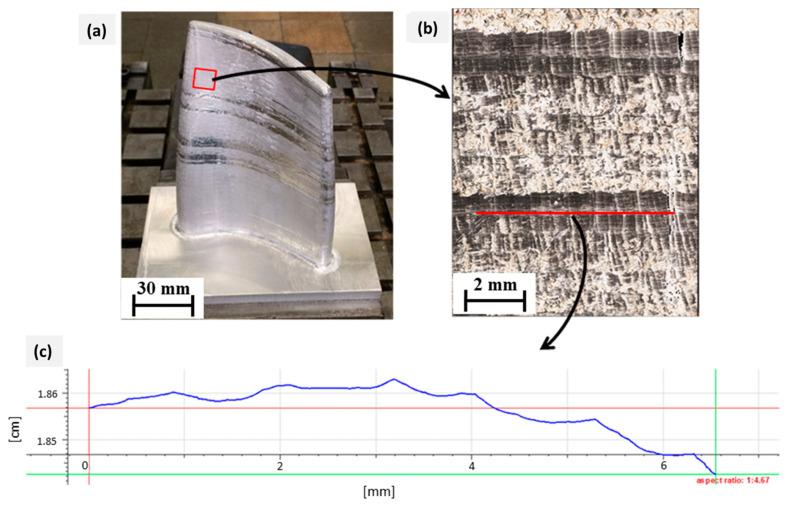
Illustration of prototype (**a**) finished after second phase of milling, (**b**) higher magnification of machined surface, and (**c**) topographical analysis in the direction of tool feed movement.

**Figure 25 materials-15-08631-f025:**
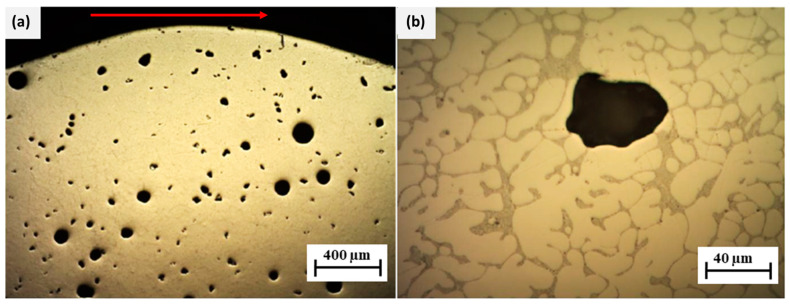
Optical microscopy of prototype cross-section wall with 11 layers (**a**) at 50× magnification, and (**b**) site with porosity and dendritic structure visible at 500× magnification. Red arrow inset in (**a**) shows the direction of cutting of the WAAM cross-section of AlSi5 blade piece.

**Figure 26 materials-15-08631-f026:**
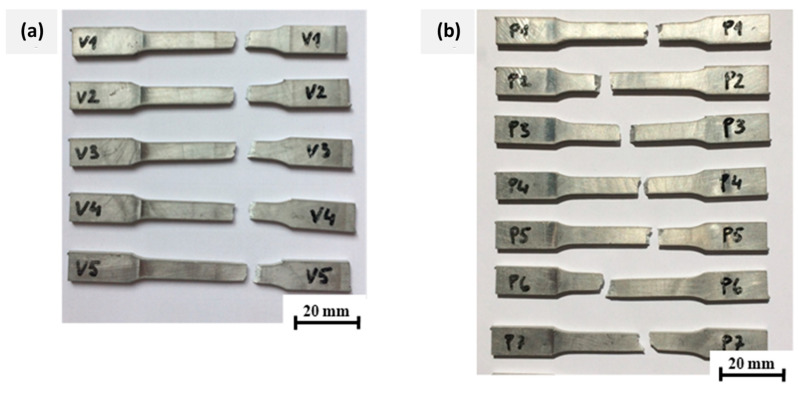
Tensile test-fractured specimens: (**a**) 5 with longitudinal angle of 0° with respect to base orientation of the layers, and (**b**) 7 with transverse orientation of 90° of welding layers with respect to base.

**Figure 27 materials-15-08631-f027:**
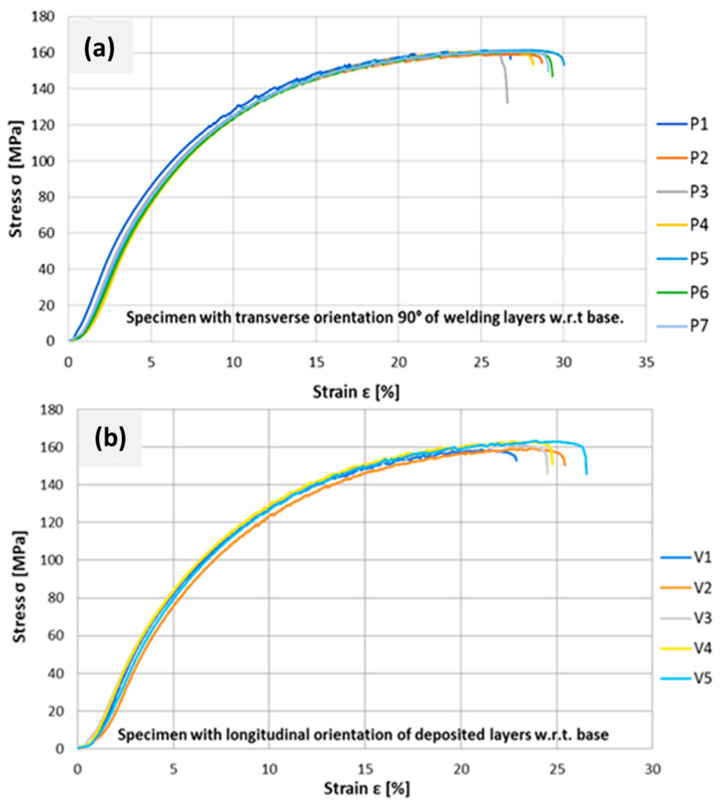
The stress–strain graph illustrates the elongation of the test pieces (**a**) in the transverse direction to the welding direction, and (**b**) in longitudinal to welding directions.

**Figure 28 materials-15-08631-f028:**
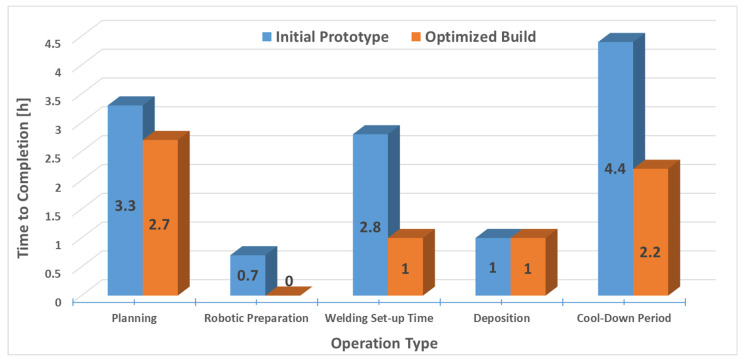
The time distribution histogram for the weldment-processing operations.

**Figure 29 materials-15-08631-f029:**
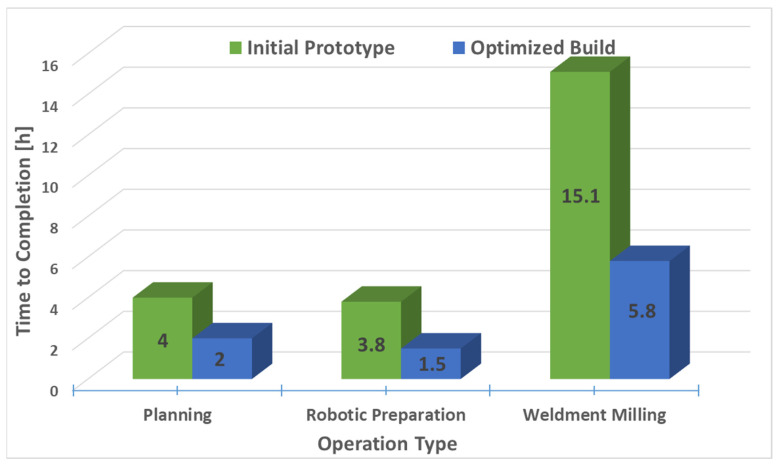
The time distribution histogram for the milling operations.

**Table 1 materials-15-08631-t001:** Physical Properties and composition of AlSi5 filler material (EN 18273: 4043) used in wire deposition [7].

**Density [kg/m^3^]**	2680
**Melting Point [°C]**	537–625
**Tensile Strength [MPa]**	120–165
**Plasticity limit [MPa]**	20–40
**Elongation [%]**	15–25

Chemical composition [wt.%]: Si = 4.5–6, Fe ≤ 0.6, Cu ≤ 0.3, Mn ≤ 0.15, Zn ≤ 0.1, Ti ≤ 0.15, Be = 0.0003, and Al = Balance.

**Table 2 materials-15-08631-t002:** Combination of welding parameters yielding different types of deposition characteristics for WAAM optimization.

Sample	Welding Speed, *v*_0_ [mm/s]	Current Strength, *I_W_* [A]	Welding Voltage, *U* [V]	Wire Feed Rate, *v_w_* [m/min]	Wall Thickness, *d_w_* [mm]	Avg. Layer Height, *h_avg_* [mm]	Remarks
A1	10	96	12.7	5.3	6.2	1.6	Corrugated Wall
A2	10	80	12.4	4.7	5.6	1.8	Corrugated Wall
A3	10	59	11.5	3.7	4.3	1.7	Flat wall
A4	10	40	10.8	2.6	–	–	Poor Remelting
B1	8	96	12.7	5.3	6.8	1.6	Sitting Ends
B2	8	80	12.4	4.7	6.0	2.0	Wavy Wall, Sowing End
B3	8	59	11.5	3.7	4.6	1.9	Flat Wall, Even Ends
B4	8	40	10.8	2.6	3.4	1.6	Poor Remelting
**B5**	**8**	**73**	**12.1**	**4.1**	**5.2**	**1.8**	**Flat Wall, No Settling**
C1	6	96	12.7	5.3	8.5	2.2	Sitting Ends
C2	6	80	12.4	4.7	6.8	2.2	Sitting Ends
C3	6	59	11.5	3.7	5.7	2.2	Corrugated Wall
C4	6	40	10.8	2.6	4.8	1.8	Poor Remelting

**Table 3 materials-15-08631-t003:** Experimental WAAM parameters for material deposition [7].

**Welding Current *I* [A]**	72–76
**Welding Voltage *U* [V]**	11.8–12.1
**Wire feed rate *v_w_* [m/min]**	4.1
**Torch Feed Rate *v*_0_ [mm/s]**	8.0
**Shielding gas flow rate** $\dot{V_{p}}$ **(L/min)**	13.0–13.5
**Initial Welding Current *I_s_* [A]**	100 (I · 135%)
**Initial Current Duration *t_s_* [s]**	0.2
**Initial Transition Time *Sl*_1_ [s]**	0.2
**Final Transition Time *Sl*_2_ [s]**	0.2
**Duration of Final Current *t_e_* [s]**	0.1
**Final Welding Current *I_e_* [A]**	36.5 (I · 50%)

**Table 4 materials-15-08631-t004:** Milling parameters of the blade prototype.

Processing Phase	Phase I Machining	Phase II Finishing
**Cutting-Spindle Speed *n* [min^−1^]**	5500–7000	6000
**Cutting Speed *v_c_* [m/min]**	105–121	113–115
**Milling Depth *a_p_* [mm]**	0.2–0.5	0.3
**Milling width *a_e_* [mm]**	0.15–0.35	0.2–0.25
**Feed Rate *v_f_* [mm/min]**	480–700	600
**Feed per tooth *f_z_* [mm/tooth]**	Depending on parameters	0.05
**Side-Cutting Tilt Angle *β* [°]**	30	30
**Tool-Clamping Length *L* [mm]**	65	65

**Table 5 materials-15-08631-t005:** Measurements of weld wall geometry and mass of deposited alloy.

Layer Number	Deposit Height [mm]	Average Section Height [mm]	Number of Layers	Average Layer Height *h_avg_* [mm]	Average Layer Thickness *d_s_* [mm]	Mass of Alloy Deposited *m* (g)
1	0–20	23.5	13	1.8	6.0	94.6
2	20–40	22.5	13	1.7	5.6	89.7
3	40–60	22.9	13	1.8	5.5	90.4
4	60–80	22.4	13	1.7	5.8	86.1
5	80–100	22.6	13	1.7	5.7	94.5
6	100–120	21.0	12	1.8	5.7	74.0
7	120–140	21.4	13	1.6	5.5	89.9
8	140–153	16.0	10	1.6	/	63.1

**Table 6 materials-15-08631-t006:** Surface roughness parameters after surface deposition and finishing by milling.

Surface Roughness Parameter	After Weld Deposition	After Phase 2 Milling
Average height of selected area *S_a_* [µm]	36	14.3
Maximum height of protrusions in selected area *S_P_* [µm]	149.6	82.2
Maximum depth of depressions in selected area *S_V_* [µm]	200.5	93.3
Maximum height of selected area *S_Z_* [µm]	350.1	175.5

**Table 7 materials-15-08631-t007:** Results of tensile tests of specimens at an angle of 0° and 90° to the base surface.

**Sample Orientation [°]**	**YS_0.2_ [MPa]**	**UTS [MPa]**	**Strain ε [%]**
0	64.7	159.2	24.8
90	58.1	161.3	28.4

**Table 8 materials-15-08631-t008:** Overview of the costs of combined production of a test piece.

Operation		Costs of Operation per Piece (EUR)		
Build Type		Initial Prototype	Optimized Build	SLM
Filler material (Hybrid = AlSi5, SLM = Al)		6	6	44.7
Shielding gas and Base Preparation		34.6	14	220
Welding	CAPP + CAM + Robot Setup	273	148	–
	CMT MIG-Welding Deposition	218	128	–
Milling	CAM + Robotic Preparation	313	140	–
	Robotic Milling	977	232	–
SLM	Laser-Melting Fabrication	–	–	1487.5
Total Costs *S* (EUR)		1802	669	1752.2

## Data Availability

Not applicable.

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
