# Peer review of "Sustainable Hybrid Manufacturing of AlSi5 Alloy Turbine Blade Prototype by Robotic Direct Energy Layered Deposition and Subsequent Milling: An Alternative to Selective Laser Melting?"

_materials, 2022, doi:10.3390/ma15238631_

Round 1

Reviewer 1 Report

What is new being presented in this article? Put this information in the abstract.

Wherever applicable, the scientific explanation needs to be added and the research novelties need to be clearly emphasized.

The gap area in the research is not clear.

At the end of Introduction section, it would be better to add the paper's organization in different sections.

Further, results and analysis of experiments should be compared with previous researchers by citing references

Improve the conclusion.

Please check the manuscript for wrong choice of words, grammatical errors and incoherent sentence structure.

Author Response

Please find the changes in letter attached.

Thank you.

Reviewer 2 Report

The authors presented an article « An elaborative approach to sustainable hybrid manufacturing of AlSi5 alloy turbine blade prototype by robotic direct energy layered deposition and subsequent milling». The article is interesting, the content is very rich and in line with the goals of “Materials”. However, there are several points in the article that require further explanation before acceptance for publication.

Title needs to be concretized. What exactly is explored in the article? By what methods? Also, it is current form is too long and confusing, please properly revise it.  

Keywords. It can exceed the maximum allowed for the journal. Please recheck it.

The abstract needs to be improved. Demonstrate in the abstract novelty, practical significance. Add more quantitative and qualitative work results to the abstract.

Each one of the cited references  must be discussed individually and demonstrate their significance to your work. Not [2-7], should be [2] text what is presented in the manuscript [2] text what is presented in the manuscript [7]. Maybe you should decrease numbers of references.

There is an interesting approach and design exists, I just propose to emphasis the practical significance of the presented methodology in several points of article.

Indeed, there are an impressive amount of results. Especially, results part of the manuscript is well enough and highly satisfying. However, the conclusions section needs to improve with selected and highlighted main findings. In conclusion section, it is necessary to more clearly show the novelty of the article and the advantages of the proposed method. Add qualitative and quantitative results of your work. Please try to emphasize your novelty, put some quantifications, and comment on the limitations. This is a very common way to write conclusions for a learned academic journal. The conclusions should highlight the novelty and advance in understanding presented in the work.

Language used in the manuscript is generally satisfying. However, writers should pay more attention of singular / plural nouns. Also, they should control the spell check/ punctuation of words and sentences. Please check all manuscript for language and misspellings. Also, please recheck upper and lower case letter. In addition, spaces should be added between words and numbers. Please revisit all manuscript and correct such inconsistencies.

There is a reference problem. First, your reference list contains any paper from “Materials” journal. If your work is convenient for this journal’s context then there are many references from this source. Second, the reference list needs to be revised and 50% of the citations must be published in the last 5 years regarding of your manuscript’s context.

References are not enough. Such a work deserves many citations. Minimum 10-15 references need to be added and some of them should be discussed.

All the article is too long. the reader forgets some of the results by the end. Try to shorten. Maybe move some results to Supplementary.

Author Response

Please find the requested changes in the attachment.

Thank you.

Reviewer 3 Report

Suggestions for improving the manuscript are as follows:

1. The Abstract should be corrected. The first three sentences are well-known. The abstract must be presented: problematic, objective, idea, methods, results, quantitative comparison of results with significant findings, conclusions.

2. Write a critique of previous research and highlight your scientific contribution.

3. In Figure 4, there is no need to highlight dimensions without numbers.

4. "The design of experiments (CAPP) defined in the flow chart are illustrated by Figure 1." Why do you use the abbreviation CAPP for design of experiments?

5. How WAAM parameters (table 3) and milling parameters (table 4) were selected? Why are they representative for this research?

Author Response

(The authors gave the same response as above.)

Round 2

Reviewer 1 Report

Authors have made significant changes in the revised manuscript. Hence, accept the manuscript for publication in its present form.

Reviewer 2 Report

Authors have improved the manuscript and made the corrections required. I think it can be accepted in present form. No more comments.